# Wearables As Graph: Personalized Health Insights via Dynamic Retrieval from Adaptive Knowledge Graphs

## Abstract

The growing prevalence of multi-sensor wearable devices generates vast amounts of long-term, multimodal time-series data, posing significant challenges for manual analysis and context-aware Large Language Models (LLMs). Current LLM-based health analysis methods typically rely on manually curated context, which becomes impractical with increasing data volume and sensor diversity. To overcome these limitations, we introduce **Wearable As Graph (WAG)**, a novel framework that automates context retrieval for LLMs using personalized knowledge graphs. WAG constructs knowledge graphs mapping relationships between wearable modalities and incorporates user-specific data. We develop a data-driven retrieval pipeline that leverages both global (long-term) and local (short-term) relationships within metrics to identify the most relevant nodes for user queries. We evaluate WAG on a benchmark of over 10k data-associated queries created from multiple wearable datasets. Both LLM- and human-based evaluations show that WAG substantially improves response quality, achieving a ∼70% win rate over baseline methods. Ablation studies further demonstrate the complementary value of global modeling (implemented via Hierarchical Bayesian Modeling to integrate general knowledge, population trends, and individual variation) and local modeling (adapted based on anomalies and query openness). WAG pioneers a wearable knowledge graph, a tailored retrieval algorithm, and a real-data based query set, creating a foundation for future research in wearable-based health monitoring.

## 1 Introduction

Mobile and wearable sensors have become powerful tools for collecting rich behavioral and health data. While clinical experts can analyze short-term, single-sensor data, long-term and multimodal analysis presents significant challenges due to human cognitive limitations. Large Language Models (LLMs) have demonstrated remarkable capabilities in interpreting time-series data—whether as raw values, reprogrammed patches, or encoder embeddings—often surpassing specialized models in pattern recognition tasks (Jin et al., 2024; Chan et al., 2024; Zhou et al., 2022; Mo et al., 2024; Gruver et al., 2023). Researchers have successfully applied LLMs to wearable data, combining textual and temporal information for health predictions across domains such as sleep, activity (Kim et al., 2024; Liu et al., 2023; Merrill et al., 2024; Englhardt et al., 2024), nutrition (Sempionatto et al., 2021), and mental health (Tazarv et al., 2021; Vos et al., 2023; Salekin et al., 2018).

Despite these advances, most existing methods require manual context preparation tailored to specific tasks. As wearable devices incorporate more sensors and accumulate longer time series, providing all data as context to LLMs becomes infeasible. Longer contexts increase computational cost, inference time, and the risk of including irrelevant information, which can reduce analytical performance (Liu et al., 2024).

To address these challenges, we propose **Wearable As Graph (WAG)**: a context retrieval framework that enables LLMs to automatically identify and retrieve relevant sensor data based on user queries. Building on the established use of knowledge graphs in medical domains such as Electronic Health Records (EHRs), WAG also employs a graph-based Retrieval-Augmented Generation (RAG) process. This method integrates and aligns multimodal sensor data, retrieving the most informative context to support robust, evidence-based LLM analysis.

Our main contributions are as follows:

- We introduce the first knowledge graph for wearable sensors, capturing connections across common modalities while supporting personalization with user-specific data.
- We construct a query set of over 10k data-associated queries from multiple wearable datasets to benchmark our framework.
- We design a data-driven context retrieval pipeline that combines long-term relationships (global modeling) with short-term anomalies (local modeling) of metrics to enhance health analysis using LLMs.
- We conduct both LLM- and human-based evaluations. Results show that WAG achieves a 70% win rate over baselines. Ablation studies validate the effectiveness of global modeling (via Hierarchical Bayesian Modeling, integrating general knowledge, population trends, and individual variations) and local modeling (capturing anomalies and balancing exploratory vs. conservative reasoning based on query openness). Human evaluations, though with high inter-rater reliability, also align closely with LLM findings.

## 2 RELATED WORK

**LLM for Wearable Sensing** Large Language Models (LLMs) have shown strong capabilities in interpreting time-series data.(Gruver et al., 2023; Jin et al., 2024) Their zero-shot reasoning ability has spurred widespread use in automated data analysis,(Chakraborty et al., 2024; Guo et al., 2024; Hong et al., 2024; Jiang et al., 2023; Hegselmann et al., 2023) where time-series signals are especially common in wearable health sensing.(Tazarv et al., 2021; Vos et al., 2023; Salekin et al., 2018; Belyaeva et al., 2023) Integrating LLMs into this domain holds promise not only for improving prediction and forecasting but also for generating meaningful insights that extend beyond label outputs.(Kim et al., 2024; Liu et al., 2023; Merrill et al., 2024; Englhardt et al., 2024; Ma et al., 2023; Strömel et al., 2024; Choe et al., 2015) However, existing methods typically assume that all relevant data is readily available. Our work addresses this gap by introducing an automated context retrieval process that selects suitable health data from large clusters of sensor signals based on user queries, prior to downstream analysis.

**Graph-based RAG** Retrieval-Augmented Generation (RAG)(Lewis et al., 2020) equips LLMs with external knowledge, offering an efficient alternative to retraining.(Hu et al., 2022) Graphs, structured representations of concept relationships, are widely used as knowledge bases to improve LLM reasoning.(Sun et al., 2018; Rotmensch et al., 2017; Edge et al., 2025) Although LLMs encode broad medical knowledge,(Singhal et al., 2023a) they often fall short in delivering contextualized analyses in applications such as electronic health record (EHR) analysis (Shi et al., 2024; Kweon et al., 2024; Liu et al., 2022; Cui et al., 2024; Choi et al., 2018; Yang et al., 2022) and medical question answering (QA).(Tang et al., 2024; Toma et al., 2023; Tu et al., 2024; Singhal et al., 2023b; Saab et al., 2024) To bridge this gap, graph-based RAG methods have been explored for injecting precise, in-domain medical knowledge.(Fei et al., 2021; Bhoi et al., 2021; Chen et al., 2019; Shang et al., 2019; Jiang et al., 2025) Wearable data analysis presents a similar challenge, as it may also requires expertise-level health knowledge that LLMs may lack. Yet, to the best of our knowledge, no knowledge graph currently captures the connections among wearable health metrics. This gap motivates our development of such a graph to enable more effective, contextualized analysis of wearable health data.

## 3 METHOD

WAG is designed to construct a personalized knowledge graph (PKG) that stores both general knowledge and user-specific wearable data. Through carefully designed graph-based retrieval, WAG leverages the PKG to provide richer and more context-aware health insights. The framework consists of four key stages: (1) query set construction, (2) knowledge graph construction, (3) query inference using a personal knowledge graph (PKG), and (4) evaluation.

### 3.1 QUERY SET CONSTRUCTION

To simulate the construction of PKGs and the querying process, we used existing wearable datasets that record various daily health metrics across multiple participants. These datasets enabled us

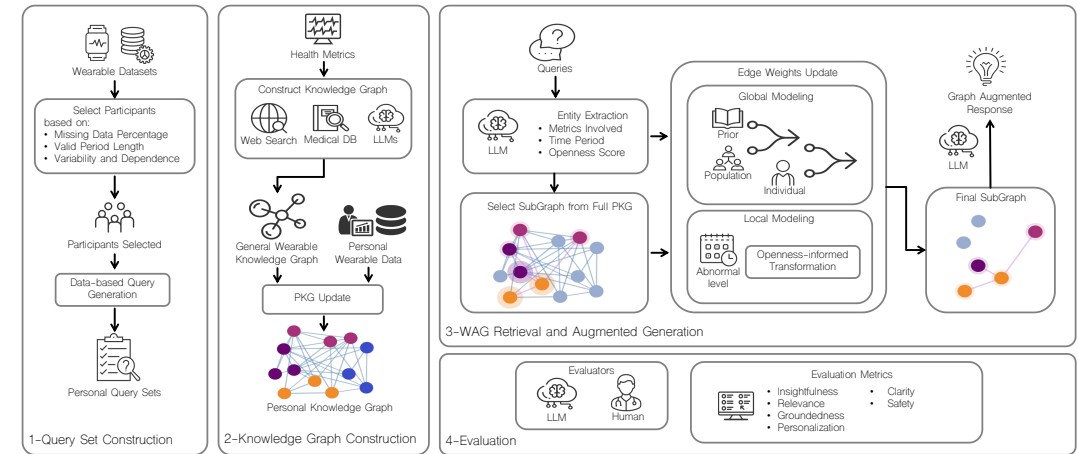

Figure 1: Main diagram.

to generate a diverse data associated query set reflecting single- and multi-metric questions. By leveraging real-world data, our simulation closely mirrors practical scenarios in which WAG would provide personalized, context-aware insights.

PARTICIPANT SELECTION

For a given dataset $\mathcal{D}$, we selected a subset of subjects $\mathcal{S}_{\text{sel}} \subset \mathcal{S}$. The selection was based on the following criteria to ensure a diverse and representative sample:

**Missing Data Percentage**: The percentage of missing data for a subject $s$ is calculated across all metrics $m \in \mathcal{M}^{\mathcal{D}}$ as:

$\text{MD}_s = \frac{1}{|\mathcal{M}^{\mathcal{D}}|} \sum_{m \in \mathcal{M}^{\mathcal{D}}} \frac{|\{t \in \mathcal{T}_s : v_{s,m,t} = \emptyset\}|}{|\mathcal{T}_s|}$, where $\mathcal{M}^{\mathcal{D}}$ is the set of all measured metrics, $\mathcal{T}_s$ is the set of all timestamps for participant $s$, and $v_{s,m,t}$ is the value of metric $m$ at time $t$.

**Valid Period Length**: The length of a participant's data collection period is defined as: $\text{VL}_s = \max(\mathcal{T}_s) - \min(\mathcal{T}_s)$

**Data Variability and Interdependence**:

- *Coefficient of Variation*: The overall variability of a participant's data across all metrics: $\text{CV}_s = \sum_{m \in \mathcal{M}_{\mathcal{D}}} \frac{\sigma_{s,m}}{\mu_{s,m}}$, where $\sigma_{s,m}$ and $\mu_{s,m}$ are the standard deviation and mean, respectively, of metric $m$ for subject $s$.

- *Pairwise Mutual Information*: The total pairwise mutual information between all metrics, quantifying their statistical dependencies: $\text{MI}_p = \sum_{\substack{(m_i, m_j) \in \mathcal{M}^{\mathcal{D}2} \\ i < j}} I(m_i; m_j)$

Participants were first selected based on high data completeness and recording duration. We then applied stratified sampling across deciles of data variability, ensuring that the final cohort represents a wide range of physiological dynamics and data conditions.

Then, for each selected participant, we sample specific timestamps and periods of interest for various metrics. This sampled data forms the foundational evidence used to construct data-driven queries. The query generation process is divided into two branches: single-metric and multi-metric queries.

QUERY GENERATION

**Single Metric Query**  For each participant $s \in \mathcal{S}_{\text{sel}}$ and each metric $m \in \mathcal{M}^{\mathcal{D}}$, we analyze the data over a set of predefined temporal windows $\mathcal{K} = \{1 \text{ day}, 7 \text{ days}, 14 \text{ days}, 30 \text{ days}, \text{all time}\}$. For a given window size $k \in \mathcal{K}$:

- *numeric metrics*: For metric $m$, we compute the abnormal level by computing the rolling average of the absolute Z-scores of temporal window $k$: $\zeta_{s,m,t} = \frac{1}{k} \sum_{i=0}^{k-1} \left| \frac{v_{s,m,t-i} - \mu_{s,m}}{\sigma_{s,m}} \right|$.

We sample timestamps $t$ where $\zeta_{s,m,t}$ falls into one of three anomaly levels: *low* (bottom 33%), *medium* (34–66%), or *high* (top 33%). Additionally, we sample timestamps $t_{\text{missing}}$ where the original data point is missing ($v_{s,m,t} = \emptyset$).

- *non-numeric metrics (e.g., text)*: We randomly select a timestamp $t$ where a valid entry exists ($v_{s,m,t} \neq \emptyset$).

The resulting input tuple for generating a single-metric query is

$$\mathcal{I}^{\text{single}} = \big(\text{metric } m, \ \text{timestamp } t, \ \text{temporal window } k, \ \text{abnormal levels } \zeta_{s,m,t,k}\big).$$

**Multi-Metric Query** For each participant $s \in \mathcal{S}_{\text{sel}}$, multi-metric queries are generated as follows. We first randomly select a subset of metrics $\mathcal{M}_{\text{sel}} \subset \mathcal{M}^{\mathcal{D}}$ with $|\mathcal{M}_{\text{sel}}| \in \{2, 3\}$. Next, we identify a timestamp $t$ at which all selected metrics have valid data, i.e., $\forall m \in \mathcal{M}_{\text{sel}}, \ v_{s,m,t} \neq \emptyset$. We then randomly choose a temporal window $k \in \mathcal{K}$.

The resulting input tuple for generating a multi-metric query is

$$\mathcal{I}^{\text{multiple}} = \big(\text{selected metric set } \mathcal{M}_{\text{sel}}, \ \text{timestamp } t, \ \text{temporal window } k, \ \text{abnormal levels } \zeta_{s,\mathcal{M}_{\text{sel}},t,k}\big).$$

**Query Types** We predefined a set of question categories, along with their openness ranges, for both single- and multi-metric queries, with additional details provided in Appendix Table 24. These categories span openness levels $\eta \in [0, 1]$, which quantify how open-ended or exploratory a query is.

| | |
|---|---|
| **Single-Metric** | General Knowledge (0.2–0.4), Data Retrieval (0.1–0.3), Trend Analysis (0.4–0.6), Comparative Insight (0.5–0.7), Anomaly Detection (0.6–0.8), Actionable Advice (0.3–0.5), Exploratory Analysis (0.7–1.0) |
| **Multi-metric** | Metric Relationships (0.4–0.6), Contextual Queries (0.5–0.7) |

Queries with a high openness score invite a broad range of responses, often requiring exploration of multiple contributing factors. In contrast, low-openness queries tend to be more closed-ended, eliciting direct or binary answers with limited elaboration. Importantly, phrasing can shift a query's openness even if the intent remains similar. For instance: *"Do you think I am stressed?"* → low openness (binary yes/no response). *"I am feeling stressed, do you have an idea why?"* → high openness (encourages interpretation and reasoning).

The resulting query tuples are passed to a LLM via the QUERYGEN module (Appendix; Prompt 1, Prompt 2), which takes $[\mathcal{I}_1, \mathcal{I}_2, \ldots, \mathcal{I}_n]$ as input and generates the query set $\mathcal{Q}$.

### 3.2 KNOWLEDGE GRAPH CONSTRUCTION

The objective of this step is to construct a knowledge graph $\mathcal{G} = (\mathcal{V}, \mathcal{E})$ to model the interconnections among various health-related metrics. In this graph, $\mathcal{V}$ represents the set of nodes corresponding to different health metrics, while $\mathcal{E}$ denotes the edges that capture relationships between these metrics.

**Nodes** Each node $v \in \mathcal{V}$ represents a distinct health metric and is assigned to one of seven predefined categories $c \in \mathcal{C}$:

$$\mathcal{C} = \{Physiological, Sleep, Activity, Mental, Environmental, Lifestyle, Demographic\}.$$

Each node is further characterized by the attributes *Name*, *Description*, *Range*, *Recommendations*, and *Data Source*, etc. as detailed in Appendix Table 26.

**Edges** Each edge $e_{i,j} \in \mathcal{E}$ is undirected and encodes the relationship between nodes $v_i$ and $v_j$. Each edge is characterized by the following attributes: *Relationship*, *Description*, *Weight* $w_{i,j}$, as detailed in Appendix Table 27.

**Knowledge Extraction and Processing** The textual information associated with nodes and edges—including descriptions, units, ranges, and recommendations—is initially gathered through web searches, scientific literature, and the Unified Medical Language System (UMLS). For web-based sources, only pages from a curated list of trusted domains are used to ensure reliability. If

a node pertains to personal health data, relevant contextual information, such as sensing signals, measurement devices, and specialized value ranges, is also incorporated. To enhance completeness and ensure evidence-based knowledge integration, all retrieved content is further processed using a large language model (LLM) guided by carefully crafted prompting strategies.

### GENERAL WEARABLE GRAPH

We started by introducing $\mathcal{M}^0$, a set of health metric concepts (*e.g.*, heart rate, step count) commonly measured by wearable devices, verified by medical experts. These metrics form the initial node set $\mathcal{V}^0 = (v_1, v_2, ..., v_{|\mathcal{M}^0|})$ of our graph $\mathcal{G}^0 = (\mathcal{V}^0, \mathcal{E}^0)$, where edges $\mathcal{E}^0 = (e_1, e_2, ..., e_{|\mathcal{E}^0|})$ represent pairwise relationships between nodes, with $|\mathcal{E}^0| = \binom{|\mathcal{M}^0|}{2}$. Edge weights $w$ are initialized using a predefined prior $w^{\text{prior}}$; in our setup, they are assigned by an LLM and validated by human experts.

### PERSONAL DATA INTEGRATION AND GRAPH EXTENSION

To enhance the initial general knowledge graph $\mathcal{G}^0$ with personal health metrics, we introduce a set of novel measurable quantities $\mathcal{M}^{\mathcal{D}} = \{m_1, \ldots, m_{|\mathcal{M}^{\mathcal{D}}|}\}$ derived from dataset $\mathcal{D}$ to simulate individual data streams. These metrics are incorporated following a structured process to construct the personal knowledge graph $\mathcal{G}$, as formalized in Algorithm 1.

Here, $NodeGen$ performs knowledge retrieval from trusted knowledge bases and feeds the information to the LLM (Appendix; Prompt 4) to generate node structures for each health metric concept $m$. $UpdateNode$ updates a existing node with new sensor specific information from new metric. Similarly, $EdgeGen$ retrieves relevant knowledge and feeds it to the LLM (Appendix; Prompt 5) to generate edges

---

**Algorithm 1** PKG Update

**Require:** Graph $\mathcal{G}^0 = (\mathcal{V}^0, \mathcal{E}^0)$, Metrics $\mathcal{M}_D = \{m_1, \ldots, m_{|\mathcal{M}_{\mathcal{D}}|}\}$,
  LLM-based Functions $\{NodeGen, EdgeGen, Merge, UpdateNode\}$

1: **for** each $m_i \in \mathcal{M}_D$ **do**
2:   **if** $\exists v_k \in \mathcal{V}^0 : \text{MERGE}(m_i, v_k)$ **then**
3:     $v_k \leftarrow \text{UPDATENODE}(m_i, v_k)$
4:   **else**
5:     $v_i \leftarrow \text{NODEGEN}(m_i)$
6:     $\mathcal{V}^0 \leftarrow \mathcal{V}^0 \cup \{v_i\}$
7:     **for** each $v_j \in \mathcal{V}^0$ **do**
8:       $e_{i,j} \leftarrow \text{EDGEGEN}(v_i, v_j)$
9:       $\mathcal{E}^0 \leftarrow \mathcal{E}^0 \cup e_{i,j}$
10:    **end for**
11:  **end if**
12: **end for**
13: **return** Updated graph $\mathcal{G} = (\mathcal{V}^0, \mathcal{E}^0)$

---

between connected nodes. Finally, $Merge$ (Appendix; Prompt 6) identifies potential duplicate metrics using the LLM to prevent graph inflation from redundant nodes, ensuring that only genuinely new metrics result in new nodes. Further deatils can be found in Appendix B.1.

## 4 WAG RETRIEVING AND AUGMENTED GENERATION

For each participant $s \in S_{\text{sel}}$, given a query $q$, a large language model extracts structured components:

$$(\mathcal{M}^q, k^q, t^q, \eta^q) = QueryParse(q),$$

where $\mathcal{M}^q = \{m_1, \ldots, m_{|\mathcal{M}^q|}\}$ are detected entities or metrics, $k^q$ is the relevant time window, the reference timestamp $t^q$, and the openness score $\eta^q$.

The openness score $\eta^q$ governs two aspects of retrieval from the personal knowledge graph $\mathcal{G} = (\mathcal{V}, \mathcal{E})$: (1) the **breadth of expansion**, i.e., how many neighbors are retrieved around primary entities; and (2) the **edge weight fusion**, i.e., blending long-term (global) and short-term (local) relationship strengths. The procedure is summarized in Algorithm 2.

Each $m \in \mathcal{M}^q$ is matched to nodes in $\mathcal{V}$ using a semantic similarity function $\text{sim}(\cdot, \cdot)$ with threshold $\delta$. The resulting primary nodes $\mathcal{V}_p$ define neighborhoods $Y = \{y_1, \ldots, y_{|Y|}\}$ around each $x \in \mathcal{V}_p$.

For each neighborhood, edges are reweighted by combining global and local components:

$$w_{x,y}^{\text{final}} = (1 - \beta) \, w_{x,y}^{\text{global}} + \beta \, w_{x,y}^{\text{local}}, \quad \beta \in [0, 1], \tag{1}$$

where $w^{\text{global}} \in \mathcal{W}^{global}$ is the Bayesian-updated global weight, $w^{\text{local}} \in \mathcal{W}^{local}$ the openness-modulated local weight (defined below) and $\beta$ is the hyperparameter controlling $w^{global}$ and $w^{local}$.

GLOBAL MODELING OF LONG-TERM RELATIONSHIP MODELING

Formally, for subject $s$, the latent vector of long-term edge weights is $\Theta_x^s = [\theta_{x,y_1}^s, \ldots, \theta_{x,y_{|Y|}}^s]^\top$ estimated using a hierarchical Bayesian model (HBM) that integrates three information sources:

$$\mathcal{W}^{\text{global}} = \text{HBM}(\mathcal{W}^{\text{prior}}, \mathcal{W}^{\text{pop}}, \mathcal{W}^{\text{ind}}),$$

where $\mathcal{W}^{\text{prior}}$ follows the **Prior Distribution**: $\Theta_x^s \sim \mathcal{N}\left(\boldsymbol{\mu}_x^{\text{prior}}, \boldsymbol{\Sigma}_x^{\text{prior}}\right)$, initialized as a Gaussian prior representing general knowledge.

$\mathcal{W}^{\text{pop}}$ is the **Population Likelihood**: $R_x^{\text{pop}} \mid \Theta_x^s \sim \mathcal{N}\left(\Theta_x^s, \boldsymbol{V}_x^{\text{pop}}\right)$, representing observed relationship patterns shared across subjects.

$\mathcal{W}^{\text{ind}}$ is the **Individual Likelihood**: $R_x^s \mid \Theta_x^s \sim \mathcal{N}\left(\Theta_x^s, \boldsymbol{V}_x^s\right)$, representing observed relationship patterns specific to the individual user.

From Bayes' theorem, the full posterior distribution, combining all sources of information can be viewed as updating the population-informed posterior with the individual's data:

$$p(\Theta_x^s \mid R_x^{\text{pop}}, R_x^s) \propto p(R_x^s \mid \Theta_x^s)\, p(\Theta_x^s \mid R_x^{\text{pop}}).$$

STAGE 1: POPULATION-INFORMED POSTERIOR VIA GAUSSIAN CONJUGATE PRIORS

First, we update the prior with the population data:

$$\Theta_x^s \mid R_x^{\text{pop}} \sim \mathcal{N}\left(\boldsymbol{\mu}_x^{\text{pop}}, \boldsymbol{\Sigma}_x^{\text{pop}}\right),$$

$$\boldsymbol{\Sigma}_x^{\text{pop}} = \left((\boldsymbol{\Sigma}_x^{\text{prior}})^{-1} + (\boldsymbol{V}_x^{\text{pop}})^{-1}\right)^{-1}, \quad \boldsymbol{\mu}_x^{\text{pop}} = \boldsymbol{\Sigma}_x^{\text{pop}}\left((\boldsymbol{\Sigma}_x^{\text{prior}})^{-1}\boldsymbol{\mu}_x^{\text{prior}} + (\boldsymbol{V}_x^{\text{pop}})^{-1}R_x^{\text{pop}}\right)$$

STAGE 2: SUBJECT-SPECIFIC POSTERIOR (FINAL)

We then update the population-informed posterior with the individual's data:

$$\Theta_x^s \mid R_x^{\text{pop}}, R_x^s \sim \mathcal{N}\left(\boldsymbol{\mu}_x^s, \boldsymbol{\Sigma}_x^s\right),$$

$$\boldsymbol{\Sigma}_x^s = \left((\boldsymbol{\Sigma}_x^{\text{pop}})^{-1} + (V_x^s)^{-1}\right)^{-1}, \quad \boldsymbol{\mu}_x^s = \boldsymbol{\Sigma}_x^s\left((\boldsymbol{\Sigma}_x^{\text{pop}})^{-1}\boldsymbol{\mu}_x^{\text{pop}} + (V_x^s)^{-1}R_x^s\right)$$

Intuitively, $(\boldsymbol{\mu}_x^{\text{prior}}, \boldsymbol{\Sigma}_x^{\text{prior}})$ encode prior knowledge about the $x$–$Y$ relationship in the graph, whereas $(R_x^{\text{pop}}, V_x^{\text{pop}})$ and $(R_x^s, V_x^s)$ capture population-level and subject-specific empirical relationships, respectively. Each covariance $V$ quantifies the uncertainty associated with its corresponding domain.

LOCAL MODELING
OF SHORT-TERM RELATIONSHIP

Short-term weights capture context-sensitive relationships over the past $k^q$ days relative to query time $t^q$. For node $x$ and neighbor $y$, the normalized abnormality score is: $\zeta_y = \frac{1}{k^q}\sum_{i=0}^{k^q-1}\left|\frac{v_{y,t^q-i}-\mu_y}{\sigma_y}\right|$, where $\mu_y$ and $\sigma_y$ are the historical mean and standard deviation.

The openness-modulated transformation is:

---

**Algorithm 2** WAG Retrieval

**Require:** Personal graph $\mathcal{G} = (\mathcal{V}, \mathcal{E})$, User query $q$, LLM-based function $QueryParse$, Similarity threshold $\delta$, Max number of retrieved nodes $\kappa$
1: $(\mathcal{M}^q, t^q, \eta^q) \leftarrow QueryParse(q)$ {Extract entities, period, openness}
2: $\mathcal{V}_{\text{primary}} \leftarrow \{v \in \mathcal{V} : \exists m \in \mathcal{M}^q, \ \text{sim}(v, m) > \delta\}$
3: $\mathcal{V}_{\text{sub}} \leftarrow \mathcal{V}_{\text{primary}}, \quad \mathcal{E}_{\text{sub}} \leftarrow \emptyset$
4: **for** each primary node $v_p \in \mathcal{V}_{\text{primary}}$ **do**
5: $\quad (\mathcal{V}_{\text{nbr}}, \mathcal{E}_{\text{nbr}}) \leftarrow \text{GETNEIGHBOR}(v_p, \mathcal{G}, \text{hops} = 1)$ {1-hop neighborhood expansion}
6: $\quad \mathcal{E}_{\text{nbr}}' \leftarrow \text{UPDATEWEIGHTS}(\mathcal{E}_{\text{nbr}}, \tau^q, \eta^q)$ {Apply global and local modeling, Eq. 1}
7: $\quad \mathcal{V}_{\text{top}} \leftarrow \text{RANKNODES}(\mathcal{V}_{\text{nbr}}, \mathcal{E}_{\text{nbr}}', k = \lceil \kappa/|\mathcal{M}^q|\rceil)$ {Select top-$k$ neighbors}
8: $\quad \mathcal{V}_{\text{sub}} \leftarrow \mathcal{V}_{\text{sub}} \cup \mathcal{V}_{\text{top}}$
9: $\quad \mathcal{E}_{\text{sub}} \leftarrow \mathcal{E}_{\text{sub}} \cup \mathcal{E}_{\text{nbr}}'$
10: **end for**
11: **return** $\mathcal{G}_{\text{sub}} = (\mathcal{V}_{\text{sub}}, \mathcal{E}_{\text{sub}})$

---

$$w_{x,y}^{\text{short}} = (2\eta^q - 1)\,\zeta_y + (1 - \eta^q),$$

The parameter $\eta^q$ acts as a dial between different behaviors:

- $\eta^q \approx 0$: $w_{x,y}^{\text{short}} \approx 1 - \zeta_y$, which prioritizes neighbors with consistently low abnormality.

- $\eta^q = 0.5$: $w_{x,y}^{\text{short}} = 0.5$, which is independent of $\zeta_y$. Here, the model effectively ignores short-term abnormality and applies equal weighting across neighbors.

- $\eta^q \approx 1$: $w_{x,y}^{\text{short}} \approx \zeta_y$, which emphasizes nodes with higher abnormality scores, allowing sensitivity to transient deviations and emerging irregular patterns.

**Final Retrieval.** Top-$\kappa/|\mathcal{V}_p|$ neighbors are selected for each primary node using the fused weights $w_{x,y}^{\text{final}}$ (Eq. 1). The final subgraph $\mathcal{G}^{\text{sub}}$ consists of all primary nodes, their selected neighbors, and associated reweighted edges, which are then provided to the LLM for contextualized reasoning and response generation via Appendix Prompt 7 and 8.

Further implementation details and derivations can be found in Appendices B.2 and G.

## 5 EXPERIMENT

### DATASET

In this study, we utilize several publicly available multimodal lifelogging datasets to ensure a comprehensive analysis. The selected datasets are described below: IFH Affect (Labbaf et al., 2024), Pmdata (Thambawita et al., 2020), Lifesnaps (Yfantidou et al., 2022), and Globem (Xu et al., 2022). For each dataset $\mathcal{D}$, we selected 10 groups, comprising 40 subjects in total. We start identifying 65 health metrics(Appendix; Table 28), from these datasets, a total of 52 distinct wearable metrics are selected for incorporation into our graph. A detailed breakdown of these metrics is provided in Appendix Table 29. A visualization of our created WAG PKG is shown in

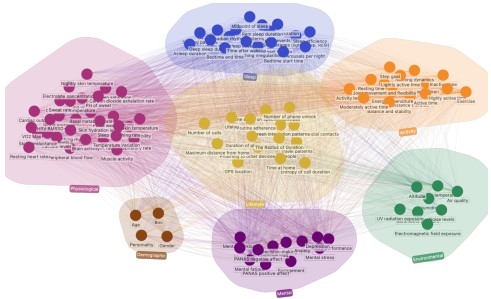

Figure 2: Visualization of the generated knowledge graph.

Figure 2. Based on these metrics and datasets, we generated a total of 10,341 unique queries.

### EVALUATION PROCEDURE

Leveraging LLMs for evaluation has proven to be an effective and scalable methodology, particularly in scenarios where standardized benchmarks are lacking (Zheng et al., 2023; Saad-Falcon et al., 2024; Chen et al., 2024; Lin & Chen, 2023). Advanced LLMs can approximate both controlled laboratory and crowdsourced human judgments, often achieving levels of inter-annotator agreement comparable to those between humans (Sottana et al., 2023; Zheng et al., 2023).

**Evaluation Metrics** Inspired from foundational concepts defined from prior work (Abbasian et al., 2024), we selected *Sensibility, Specificity, Interestingness (SSI), Groundedness, Personalization, Conciseness,* and *Safety* to formulate our own metrics:

Table 1: Evaluation dimensions for assessing response quality.

| Dimension | Description |
|---|---|
| Insightfulness | Similar to *Interestingness* in *SSI* (Thoppilan et al., 2022). Captures whether incorporating high-quality, context-aware information leads to more insightful responses. |
| Relevance | Derived from *Specificity* and *Sensibility* in *SSI*. Assesses whether the system retrieves highly relevant content tailored to the user's context. |
| Groundedness | Evaluates whether responses are supported by factual or retrievable content. |
| Personalization | Measures how accurately responses reflect the user's specific data and context. |
| Clarity | Related to *Conciseness*. Judges whether responses are clear, and accessible, even for complex queries. |
| Safety & Security | Ensures responses avoid unsafe, harmful, or misleading content. |
| Overall Quality | A holistic assessment combining the above dimensions to capture the overall usefulness and reliability of responses. |

Our evaluation procedure is as follows: For a given query, the responses from all evaluated methods are presented to a powerful LLM judge. The judge is instructed to rank the responses according to

Table 2: Main experiment - Comparison of our method with baselines across all datasets. Note: For rank metrics (↓), lower values are better from 1 to 3. For win rate (↑), higher values are better.

| Dataset | Method | Rank ↓ | | | | | | | Overall Win Rate↑ |
|---|---|---|---|---|---|---|---|---|---|
| | | Insight | Relevance | Grounded | Personal | Clarity | Safety | Overall | |
| Globem | Base | 2.57 | 2.57 | 2.42 | 2.55 | 2.24 | 1.00 | 2.57 | 0.06 |
| | Rag | 2.00 | 1.99 | 2.03 | 2.00 | 1.86 | 1.00 | 1.99 | 0.24 |
| | WAG | **1.43** | **1.44** | **1.50** | **1.44** | 1.82 | 1.00 | **1.43** | **0.70** |
| IFH Affect | Base | 2.59 | 2.58 | 2.45 | 2.57 | 2.24 | 1.00 | 2.59 | 0.06 |
| | Rag | 2.01 | 2.01 | 2.05 | 2.02 | 1.90 | 1.00 | 2.01 | 0.23 |
| | WAG | **1.40** | **1.40** | **1.46** | **1.41** | 1.79 | 1.00 | **1.40** | **0.71** |
| Lifesnap | Base | 2.66 | 2.65 | 2.50 | 2.63 | 2.31 | 1.00 | 2.66 | 0.05 |
| | Rag | 1.95 | 1.95 | 2.00 | 1.96 | 1.83 | 1.00 | 1.95 | 0.25 |
| | WAG | **1.39** | **1.40** | **1.45** | **1.40** | 1.75 | 1.00 | **1.39** | **0.70** |
| Pmdata | Base | 2.63 | 2.61 | 2.45 | 2.60 | 2.28 | 1.00 | 2.62 | 0.06 |
| | Rag | 1.97 | 1.97 | 2.04 | 1.98 | 1.85 | 1.00 | 1.96 | 0.23 |
| | WAG | **1.41** | **1.41** | **1.47** | **1.41** | 1.78 | 1.00 | **1.41** | **0.70** |

the criteria defined above using the structured prompt detailed in Appendix Prompt 9. The resulting rankings are then aggregated across the entire query set. We report the average rank for each method and compute the *win rate*, defined as the percentage of queries for which a system's response is ranked highest. To validate the reliability of the LLM-based judgments, we also perform a human evaluation on a randomly sampled subset of queries. Domain experts are asked to provide rankings for the same responses, allowing us to evaluate the connection between LLM and human judgments.

We design three experiments to evaluate the effectiveness of our proposed framework. For the LLM-judged main experiment, we use the entire constructed query set, while for Exp-G and Exp-L, we select a total of 1,000 single-metric queries, with 250 drawn from each dataset. Because different conditions may sometimes produce identical retrieval results from the graph, in Exp-G and Exp-L, we restrict our selection to queries where all conditions yield distinct retrieval results. Consequently, these queries tend to have relatively high openness scores $\eta$, which generally invite more exploration. Finally, we sample 100 queries from each experiment to construct the query set for human evaluation. The evaluation is conducted by three students with medical backgrounds using a simple web-based interface (Appendix I).

MAIN EXPERIMENT

We compare three conditions: **Baseline**: the LLM is provided only with relevant personal data, without any external context (e.g., grounded knowledge). **RAG**: a standard RAG approach, where only information directly related to the primarily detected entity is retrieved. **WAG**: our method, which dynamically adjusts edge weights based on both the user's data and the openness score, enabling more context-aware and adaptive reasoning from other related nodes.

The primary results in Table 2 highlight the superiority of our approach. Compared to the Baseline, the standard RAG method achieves a substantially lower (better) average overall rank, reflecting a ∼37.5% improvement and confirming that incorporating external knowledge consistently enhances response quality. Our proposed WAG framework delivers an even greater gain, reducing the average overall rank to ∼1.4, a ∼56% improvement over standard RAG. This is further supported by a win rate of nearly 70%, showing that WAG generated the preferred response for the majority of evaluated samples. Additional analyses (Appendix; Tables 10 and 12) show that WAG's advantage is most pronounced on queries with higher abnormality metrics and those with higher openness scores, such as Trend Analysis, Comparative Insight, Anomaly Detection, and Exploratory Analysis. These results demonstrate that WAG is particularly effective for complex, open-ended analytical scenarios where dynamic and context-aware reasoning is most critical.

ABLATION STUDIES

We conduct two ablation experiments to evaluate the effectiveness of the two core components in our WAG retrieval module: global modeling (Experiment-G) and local modeling (Experiment-L) of edge weights.

Experiment-G evaluates four weighting strategies of global modeling within our Hierarchical Bayesian Model

Table 3: Exp-G - Comparison of different weighting within global modeling across datasets.

| Dataset | $\mathcal{W}^{global}$ | $\mathcal{W}^{ind}$ | $\mathcal{W}^{pop}$ | $\mathcal{W}^{prior}$ |
|---|---|---|---|---|
| Globem | **2.14** | 2.62 | 2.50 | 2.22 |
| IFH Affect | **2.08** | 2.42 | 2.40 | 2.39 |
| Lifesnap | **2.14** | 2.41 | 2.32 | 2.34 |
| Pmdata | **2.28** | 2.42 | 2.40 | 2.32 |
| Average | **2.16** | 2.47 | 2.40 | 2.32 |

(HBM). For a primary node $\mathcal{V}_p$, the weight of a neighboring node $Y$ is defined as follows: $\mathcal{W}^{\text{prior}}$ is the initial weight based on prior knowledge from the knowledge graph ($\boldsymbol{\mu}_x^{\text{prior}}$); $\mathcal{W}^{\text{pop}}(R_x^{\text{pop}})$ is derived solely from relationships in the population data; $\mathcal{W}^{ind}(R_x^s)$ is derived solely from relationships in the individual data of user $s$; and $\mathcal{W}^{\text{global}}(\boldsymbol{\mu}_x^s)$ integrates all three sources via HBM. As shown in Table 3, $\mathcal{W}^{global}$ consistently achieves the lowest average rank across all datasets, demonstrating that integrating prior, population, and individual information improves the retrieval of relevant neighboring nodes. $\mathcal{W}^{prior}$ ranks second, highlighting the value of structured knowledge-graph relationships, while single-source strategies ($\mathcal{W}^{ind}$ or $\mathcal{W}^{pop}$) perform worst. The overall ranking ($\mathcal{W}^{global} > \mathcal{W}^{prior} > \mathcal{W}^{pop} > \mathcal{W}^{ind}$) is consistent across datasets, and a Friedman test confirms the differences are statistically significant ($p = 4.52 \times 10^{-8}$).

Experiment-L evaluates the effectiveness of local modeling by comparing three conditions. The weight is determined by $\mathcal{W}^{\text{global}}$, the weight obtained from global modeling; $\mathcal{W}^{\text{local}}$, the weight obtained through local modeling; and $\mathcal{W}^{\text{final}}$, the final weight after completing the full modeling framework. As shown in Table 4, $\mathcal{W}^{\text{final}}$ also achieves the best average rank across all datasets, indicating that the combined global–local modeling provides the most reliable weighting. Although the improvement is relatively modest (approximately 12% compared to the other two strategies), the effect is consistent across all

Table 4: Exp-L - Evaluation of the effectiveness of local modeling across datasets.

| Dataset | $\mathcal{W}^{final}$ | $\mathcal{W}^{global}$ | $\mathcal{W}^{local}$ |
|---|---|---|---|
| Globem | **1.90** | 1.98 | 2.01 |
| IFH Affect | **1.88** | 1.98 | 2.02 |
| Lifesnap | **1.85** | 1.94 | 2.10 |
| Pmdata | **1.89** | 2.01 | 1.99 |
| Average | **1.88** | 1.98 | 2.03 |

datasets. A Friedman test confirms that these differences are statistically significant ($p = 0.00151$).

HUMAN EVALUATION

Table 5: Comparison of overall ranks between human evaluators and the LLM evaluator. We report the average human rank across all evaluators and the average LLM rank across all test queries.

| Experiment | Main Exp | | | Exp-G | | | | Exp-L | | |
|---|---|---|---|---|---|---|---|---|---|---|
| Evaluator | WAG | Rag | Base | $\mathcal{W}^{global}$ | $\mathcal{W}^{ind}$ | $\mathcal{W}^{pop}$ | $\mathcal{W}^{prior}$ | $\mathcal{W}^{final}$ | $\mathcal{W}^{global}$ | $\mathcal{W}^{local}$ |
| **Human** | **1.47** | 1.90 | 2.45 | **2.31** | 2.51 | 2.46 | 2.43 | **1.92** | 1.95 | 2.05 |
| **LLM** | **1.41** | 1.98 | 2.61 | **2.16** | 2.47 | 2.40 | 2.32 | **1.88** | 1.98 | 2.03 |

As shown in Table 5, the human evaluation results are largely consistent with the trends identified by the LLM evaluator. In the main experiment, WAG is rated much higher than the other two methods, and for experiment-G, the overall ranking preference matches that of the LLM-based evaluation. For Experiment-L, the general trend is still consistent, but $\mathcal{W}^{\text{final}}$ only marginally outperforms $\mathcal{W}^{\text{global}}$, indicating some divergence between human and LLM judgments. To further investigate this, we examined the correlation between human and LLM evaluations. While the overall correlation is relatively low, this is unsurprising given the limited inter-rater reliability (IRR) among human annotators (Appendix; Table 21a). Notably, in Experiment-L (Appendix; Table 23), two of the human evaluators followed a trend similar to that of the LLM evaluators, whereas the third exhibited the opposite preference. These findings highlight both the subjectivity of the evaluation task and the challenges of achieving consistent human judgments in this setting.

We also provides some qualitative examples in Appendix J.

# 6 CONCLUSION

In this work, we introduce **Wearable As Graph (WAG)**, a graph-based context retrieval framework designed to enhance LLM-driven health analysis on wearable data. WAG integrates multimodal sensor signals into personalized knowledge graph, leveraging both global and local modeling strategies to enable LLMs to retrieve the most relevant context for diverse user queries. We also construct a query set that spans a wide range of potential user questions based on real-world wearable data, along with a general knowledge graph capturing broad domain knowledge about health and wearable metrics. Together, these resources provide a foundation for future studies in wearable-based health analysis and enable the research community to benchmark and extend context-aware LLM applications. We envision WAG as a foundational framework that can accelerate research leveraging the growing richness of wearable ecosystems.

ETHICS STATEMENT

This work focuses on methods for improving context-aware health analysis using wearable data and large language models (LLMs). While our framework, Wearable As Graph (WAG), shows promise in providing personalized insights, it is not designed or validated for direct clinical use. The datasets used in this study are publicly available and de-identified to protect participant privacy. No personally identifiable information was accessed or processed. We acknowledge that automated health analysis poses potential risks, including misinterpretation, over-reliance and biases introduced by both the underlying wearable datasets and the LLMs employed.

REPRODUCIBILITY STATEMENT

We have made every effort to provide sufficient details to enable reproduction of our results. This includes pseudocode of our proposed approach (Algorithms 1 and 2), detailed descriptions of data processing and query generation (Section 3.1), prompts (Appendix K), hyperparameters, and implementation details (Section B). All datasets used in this study are publicly available. In addition, the generated query set and code will be released to support reproducibility.

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

# Contents

## A  DATASET AND QUERYSET STATS

### A.1  DATASET

**IFH Affect** (Labbaf et al., 2024): A longitudinal dataset collected from 21 university students before, during, and after the COVID-19 lockdown in Southern California. Data was gathered over an average of 7.8 months via a Samsung Galaxy Watch, Oura Ring, the Personicle lifelogging app, and ecological momentary assessments (EMA). It includes raw sensor data (PPG, IMU), processed physiological measures (heart rate, sleep, activity), and extensive self-reported surveys on mood, mental health (BDI-II, GAD-7), and social factors, providing insights into lifestyle and emotional adjustment during major world events.

**PMData** (Thambawita et al., 2020): This dataset comprises 16 participants (12 men, 3 women, avg. age 34) monitored over 5 months. It combines objective biometrics from a Fitbit Versa 2 smartwatch with subjective self-reports collected via Google Forms (demographics, diet) and a dedicated sports logging app (PMSys) for metrics such as fatigue, mood, and stress, facilitating a link between physical activity and personal well-being.

**LifeSnaps** (Yfantidou et al., 2022): A comprehensive, multi-modal dataset from 71 participants (42 male, 29 female) collected over more than 4 months. It integrates automatically synced data from a Fitbit Sense (sleep, heart rate, stress), ecological momentary assessments (EMA) on context and mood via the SEMA3 platform, and validated surveys on demographics and health, supporting research into daily life and behavior.

**Globem** (Xu et al., 2022): A large-scale, multi-year dataset encompassing 705 user-years of data from 497 diverse participants. It was collected using the AWARE framework on mobile phones, Fitbit wearables (Flex2 and Inspire 2), and ecological momentary assessments (EMA). The dataset's scale and diversity support the study of long-term behavioral trends across a varied population.

## A.2 QUERYSET

Table 6: Statistics of query sets

| Dataset | #Queries (Exp-G) | #Queries (Exp-L) | Total #Queries (Exp-Main) |
|---|---|---|---|
| Globem | 250 | 250 | 1961 |
| IFH Affect | 250 | 250 | 2921 |
| Lifesnap | 250 | 250 | 2972 |
| Pmdata | 250 | 250 | 2487 |
| Total | 1000 | 1000 | 10341 |

Table 7: Query counts per query type

| | General Knowledge | Data Retrieval | Trend Analysis | Comparative Insight | Anomaly Detection | Actionable Advice | Exploratory Analysis | Metric Relationships | Contextual Queries |
|---|---|---|---|---|---|---|---|---|---|
| **Main experiment** | | | | | | | | | |
| Globem | 104 | 531 | 219 | 196 | 199 | 125 | 387 | 147 | 53 |
| IFH Affect | 164 | 749 | 348 | 330 | 305 | 256 | 569 | 141 | 59 |
| Lifesnap | 188 | 800 | 335 | 315 | 307 | 254 | 573 | 143 | 57 |
| Pmdata | 126 | 654 | 287 | 265 | 263 | 188 | 504 | 128 | 70 |
| Total | 582 | 2734 | 1189 | 1106 | 1074 | 823 | 2033 | 559 | 239 |
| **Experiment-G** | | | | | | | | | |
| Globem | 0 | 1 | 38 | 49 | 50 | 24 | 88 | | |
| IFH Affect | 2 | 10 | 18 | 45 | 49 | 24 | 102 | | |
| Lifesnap | 2 | 2 | 21 | 39 | 51 | 35 | 100 | | |
| Pmdata | 0 | 0 | 22 | 31 | 51 | 26 | 120 | | |
| Total | 4 | 13 | 99 | 164 | 201 | 109 | 410 | | |
| **Experiment-L** | | | | | | | | | |
| Globem | 1 | 9 | 14 | 51 | 63 | 18 | 94 | | |
| IFH Affect | 9 | 15 | 8 | 51 | 56 | 22 | 89 | | |
| Lifesnap | 13 | 13 | 16 | 39 | 50 | 22 | 97 | | |
| Pmdata | 4 | 11 | 12 | 47 | 62 | 20 | 94 | | |
| Total | 27 | 48 | 50 | 188 | 231 | 82 | 374 | | |

Table 8: Query counts per query time period

| Query Period | 1 | 7 | 14 | 30 | all |
|---|---|---|---|---|---|
| **Main Experiment** | | | | | |
| Globem | 824 | 223 | 250 | 416 | 248 |
| IFH Affect | 1028 | 373 | 468 | 763 | 289 |
| Lifesnap | 1117 | 433 | 366 | 748 | 308 |
| Pmdata | 876 | 371 | 362 | 628 | 248 |
| Total | 3845 | 1400 | 1446 | 2555 | 1093 |
| **Experiment-G** | | | | | |
| Globem | 100 | 37 | 32 | 81 | |
| IFH Affect | 79 | 34 | 43 | 94 | |
| Lifesnap | 88 | 31 | 37 | 94 | |
| Pmdata | 95 | 36 | 41 | 78 | |
| Total | 362 | 138 | 153 | 347 | |
| **Experiment-L** | | | | | |
| Globem | 114 | 32 | 36 | 68 | |
| IFH Affect | 87 | 38 | 53 | 72 | |
| Lifesnap | 97 | 50 | 36 | 67 | |
| Pmdata | 87 | 47 | 37 | 79 | |
| Total | 385 | 167 | 162 | 286 | |

Table 9: Query counts per abnormal level

| Abnormal Level | Low | Medium | High | Other |
|---|---|---|---|---|
| **Main Experiment** | | | | |
| Globem | 431 | 431 | 442 | 657 |
| IFH Affect | 722 | 722 | 734 | 743 |
| Lifesnap | 712 | 712 | 729 | 819 |
| Pmdata | 601 | 601 | 603 | 680 |
| Total | 2466 | 2466 | 2508 | 2899 |
| **Experiment-G** | | | | |
| Globem | 26 | 153 | 153 | |
| IFH Affect | 25 | 124 | 184 | |
| Lifesnap | 20 | 125 | 188 | |
| Pmdata | 9 | 120 | 204 | |
| Total | 81 | 522 | 729 | |
| **Experiment-L** | | | | |
| Globem | 14 | 108 | 128 | |
| IFH Affect | 31 | 112 | 107 | |
| Lifesnap | 30 | 94 | 126 | |
| Pmdata | 20 | 107 | 123 | |
| Total | 95 | 421 | 484 | |

## B  IMPLEMENTATION DETAILS

### B.1  GRAPH CONSTRUCTION

We provide contextual information to the LLM at every sub-stage of the graph construction process. Starting with a predefined list of metrics, we first collect relevant knowledge from trusted medical databases (e.g., UMLS) and web sources (via the Google Serper API). To ensure reliability, searches are limited to a set of verified domains.

Metric information is fed in batches to the LLM to generate the corresponding nodes. After node generation, we similarly collect knowledge for every pair of nodes to identify appropriate sources, and batched edge information is then used by the LLM to generate edges and assign weights.

During PKG construction, each candidate metric is compared against existing nodes. If it already exists, the node is updated; otherwise, a new node is created. Edges are then established between the new node and all existing nodes using the same batch-wise LLM procedure.

### B.2  GRAPH RETRIEVAL

We conducted all experiments using DeepSeek-V3. The default relevant time window $k^q$ is set to 7 days, if it is not specified in the query, and the maximum number of related nodes retrieved per query is set to $\kappa = 5$. The hyperparameter $\beta$ is set to 0.5 to balance global and local contributions in edge weighting.

The similarity function $\text{sim}(\cdot, \cdot)$ follows a standard embedding-based retrieval mechanism used in RAG frameworks and is computed via cosine similarity in the embedding space. Specifically, we compute embeddings of entity names and compare them with embeddings of node names in the graph. Through experimentation, a threshold of $\delta = 0.85$ was found to reasonably balance node hit rate and retrieval consistency. Population-level and subject-specific relationships, $R_x^{\text{pop}}$ and $R_x^s$, can be represented in different ways, such as correlation or mutual information. In our current setup, we adopt Spearman correlation because it requires less data to yield valid estimates. In contrast, mutual information generally demands much larger sample size but can capture more complex dependencies, particularly within a single user's personal data. As illustrated in Figure 3, while the relationships captured by mutual information and Spearman correlation are often consistent, estimating mutual information can be challenging when data are limited. We also favor Spearman correlation over Pearson correlation, as it better handles non-linear monotonic relationships commonly observed in our setting.

In our current setup, we define

$$R_x^{\text{pop}} = [r_{x,y_1}^{\text{pop}}, \; r_{x,y_2}^{\text{pop}}, \; \cdots, \; r_{x,y_{|Y|}}^{\text{pop}}]^\top$$

as the Spearman correlations between historical data of $x$ and each $y \in Y$ across the dataset, and

$$R_x^s = [r_{x,y_1}^s, \; r_{x,y_2}^s, \; \cdots, \; r_{x,y_{|Y|}}^s]^\top$$

as the correlations computed from user $s$'s data. To stabilize the Gaussian modeling, we apply the Fisher $z$-transform:

$$z = \tanh^{-1}(r) = \frac{1}{2} \ln\left(\frac{1+r}{1-r}\right), \quad r = \tanh(z).$$

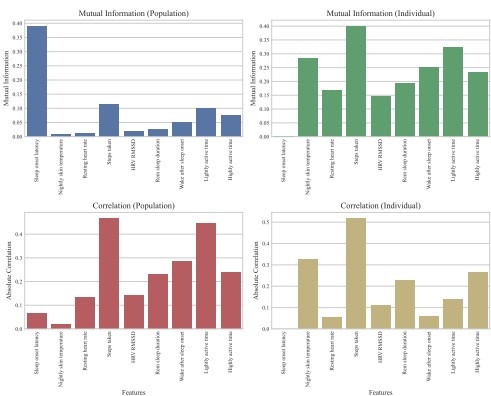

Figure 3: Comparison of weights encoded by spearman correlation and mutual information.

The covariance matrices $V_x^{\text{pop}}$ and $V_x^s$ encode the sampling variances on the $z$-scale (approximately $1/(n-3)$ for Spearman correlations) and are modulated by hyperparameters $\alpha^{\text{pop}}$ and $\alpha^{\text{ind}}$, respectively. The prior covariance is set as $\Sigma_x^{\text{prior}} = V_x^{\text{pop}}$ and is not modulated. All covariance matrices are diagonal, and we enforce a minimum of 10 samples to compute a correlation. Finally, a sigmoid function is applied to both $\mathcal{W}^{\text{global}}$ and $\mathcal{W}^{\text{local}}$ to restrict values to $[0, 1]$, with steepness hyperparameters $\gamma^{\text{global}} = 0.9$ and $\gamma^{\text{local}} = 0.7$.

For Exp-G and Exp-L, which explore different versions of WAG, we focus only on nodes with numerical data and ignore non-numerical nodes. This ensures that all relationships include both population-level and subject-specific information, allowing evaluation of the full retrieval algorithm.

**Fallback Strategy for Missing or Invalid Observations**    In practice, empirical relationships $R$ may be missing or invalid (e.g., non-numerical nodes or insufficient data). To maintain robust edge weight estimation, we employ a sequential fallback strategy:

$$w_{x,y}^{\text{final}} \propto \begin{cases} \mu_{x,y}^{s}, & \text{if } r_{x,y}^{s} \text{ is valid,} \\ \mu_{x,y}^{\text{pop}}, & \text{if } r_{x,y}^{s} \text{ is missing or invalid,} \\ w_{x,y}^{\text{prior}}, & \text{if both } r_{x,y}^{s} \text{ and } r_{x,y}^{\text{pop}} \text{ are unavailable.} \end{cases}$$

This ensures that $w^{\text{global}}$ defaults sequentially from subject-specific to population-informed to prior weights, maintaining robustness and interpretability even with incomplete data.

**Determination of Hyperparameters $\alpha^{\text{pop}}$ and $\alpha^{\text{ind}}$**    The optimal population regularization parameter $\alpha^{\text{pop}}$ was determined using a data-driven approach based on Kendall Tau similarity curves. Specifically, we compute two curves that quantify different aspects of ranking alignment:

- $\tau(\boldsymbol{\mu}^{\text{prior}}, \boldsymbol{\mu}^{\text{pop}})$ measures the preservation of the original prior ranking under increasing regularization strength.
- $\tau(R^{\text{pop}}, \boldsymbol{\mu}^{\text{pop}})$ quantifies the alignment between the regularized population posterior and observed population statistics.

The intersection of these curves identifies the value of $\alpha^{\text{pop}}$ where the regularized posterior achieves an optimal balance between faithfulness to the prior expertise and consistency with population-level preferences, avoiding both overfitting and excessive dilution of population information.

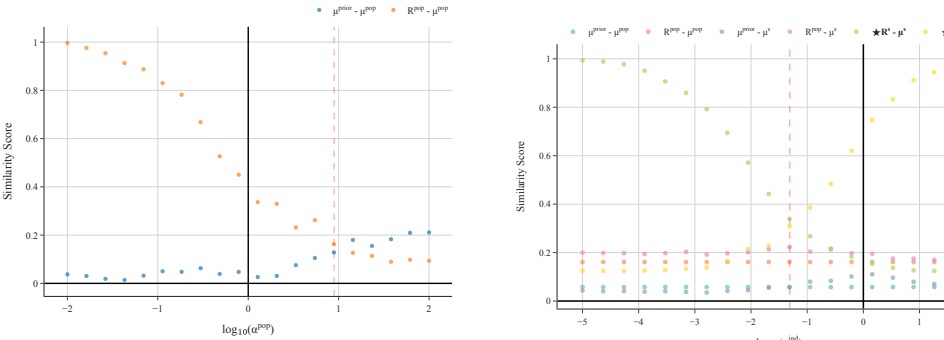

Figure 4: Kendall Tau similarity scores as a function of population regularization strength ($\alpha^{\text{pop}}$).

Figure 5: Kendall Tau similarity measures as functions of individual regularization strength ($\alpha^{\text{ind}}$).

Similarly, the individual regularization parameter $\alpha^{\text{ind}}$ is determined by analyzing the intersection of Kendall Tau similarity curves capturing individual-level alignment:

- $\tau(\boldsymbol{\mu}^{\text{pop}}, \boldsymbol{\mu}^{s})$ measures the preservation of the population posterior ranking under increasing individual regularization strength.
- $\tau(R^{s}, \boldsymbol{\mu}^{s})$ quantifies the alignment between the regularized individual posterior and empirical individual statistics.

Additional curves, such as $\tau(\boldsymbol{\mu}^{\text{prior}}, \boldsymbol{\mu}^{\text{ind}})$, $\tau(R^{\text{pop}}, \boldsymbol{\mu}^{\text{ind}})$, and $\tau(\boldsymbol{\mu}^{\text{pop}}, \boldsymbol{\mu}^{s})$, provide complementary insights into prior-individual and two-stage posterior alignment. The intersection points in these analyses identify the optimal $\alpha^{\text{ind}}$, balancing individual-specific data with population-informed priors.

## C  ILLUSTRATIVE EXAMPLE OF A SINGLE-METRIC QUERY SCENARIO

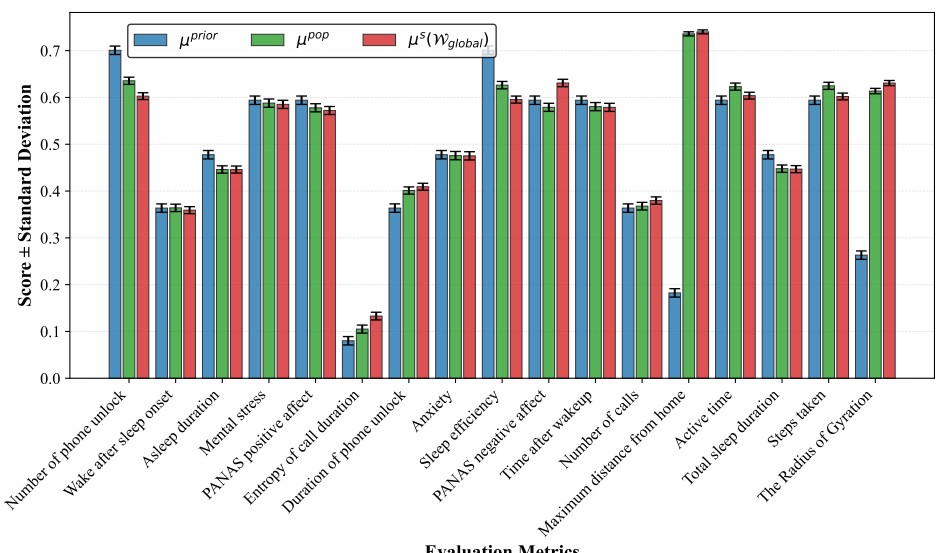

Figure 6: Visualization of hierarchical Bayesian modeling (HBM) updates for all nodes related to the metric *"Circadian rhythm patterns"*. $\boldsymbol{\mu}^{\text{prior}}$ denotes the prior, $\boldsymbol{\mu}^{\text{pop}}$ denotes the posterior after population data update, and $\boldsymbol{\mu}^s(\mathcal{W}^{\text{global}})$ denotes the final posterior after incorporating individual-specific data.

| Rank \ Weighting by | $\boldsymbol{\mu}^{prior}$ | $\boldsymbol{\mu}^{pop}$ | $\boldsymbol{\mu}^s(\mathcal{W}^{global})$ |
|---|---|---|---|
| 1 | Number of phone unlock (0.70) | Maximum distance from home (0.74) | Maximum distance from home (0.74) |
| 2 | Sleep efficiency (0.70) | Number of phone unlock (0.64) | PANAS negative affect (0.63) |
| 3 | Mental stress (0.59) | Sleep efficiency (0.63) | The Radius of Gyration (0.63) |
| 4 | PANAS positive affect (0.59) | Steps taken (0.62) | Active time (0.60) |
| 5 | Steps taken (0.59) | Active time (0.62) | Number of phone unlock (0.60) |
| 6 | PANAS negative affect (0.59) | The Radius of Gyration (0.61) | Steps taken (0.60) |
| 7 | Time after wakeup (0.59) | Mental stress (0.59) | Sleep efficiency (0.60) |
| 8 | Active time (0.59) | Time after wakeup (0.58) | Mental stress (0.59) |
| 9 | Asleep duration (0.48) | PANAS negative affect (0.58) | Time after wakeup (0.58) |
| 10 | Anxiety (0.48) | PANAS positive affect (0.58) | PANAS positive affect (0.57) |
| 11 | Total sleep duration (0.48) | Anxiety (0.48) | Anxiety (0.48) |
| 12 | Duration of phone unlock (0.36) | Total sleep duration (0.45) | Total sleep duration (0.45) |
| 13 | Number of calls (0.36) | Asleep duration (0.45) | Asleep duration (0.45) |
| 14 | Wake after sleep onset (0.36) | Duration of phone unlock (0.40) | Duration of phone unlock (0.41) |
| 15 | The Radius of Gyration (0.26) | Number of calls (0.37) | Number of calls (0.38) |
| 16 | Maximum distance from home (0.18) | Wake after sleep onset (0.36) | Wake after sleep onset (0.36) |
| 17 | Entropy of call duration (0.08) | Entropy of call duration (0.11) | Entropy of call duration (0.13) |

Figure 7: Ranking of nodes related to *"Circadian rhythm patterns"* based on different HBM weight stages. Nodes selected for retrieval are highlighted in blue.

We illustrate a single-metric query scenario using a simulated subject from the Globem dataset. Suppose the subject issues the query:

> *"What factors might be causing the significant deviations in my circadian rhythm patterns over the past 30 days?"*

The query is processed via $QueryParse$, yielding:

- Time granularity $k^q = 30$ days,
- Detected metric $\mathcal{M}^q = \{$"Circadian rhythm patterns"$\}$,
- Openness score $\eta^q = 0.8$,
- Internal reference timestamp $t^q$.

Given a predefined maximum number of related nodes $\kappa = 5$, the number of nodes retrieved is computed as: #retrieved nodes $= \eta^q \cdot \kappa = 0.8 \times 5 = 4$.

The corresponding edge weights are obtained from the subject's personal knowledge graph (PKG), combining both population-level statistics from Globem and personal data. These weights are then passed through our retrieval algorithm. Figure 6 visualizes the changes of edge weights as they are updated through the hierarchical Bayesian modeling process, showing how information flows from the prior distribution to the population-informed posterior and finally to the final posterior. Table 7 presents the ranking results derived from these weights, demonstrating that the system retrieves different related nodes depending on the weighting mechanism.

Additionallu, Figure 8 shows the distribution of various edge weights across our entire query set.

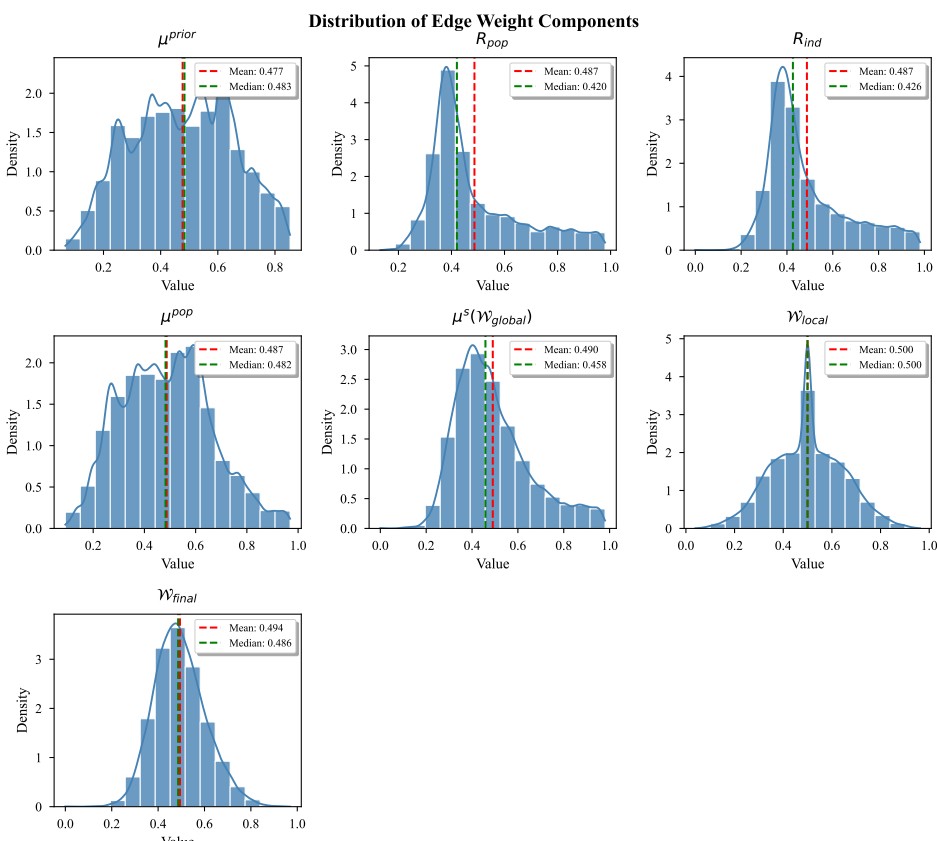

Figure 8: Distributions of different edge weight components in the personal knowledge graph. The eight subplots display: (a) prior weights ($\mu^{\text{prior}}$), representing the initial LLM-assigned weight; (b) empirical relationships from the population ($R^{\text{pop}}$), representing correlations observed across all subjects; (c) empirical relationships from the individual ($R^s$), representing correlations computed from the subject's personal data; (d) posterior after population data update ($\mu^{\text{pop}}$); (e) posterior after incorporating individual-specific data ($\mu^s(\mathcal{W}^{\text{global}})$); (f) local weights ($\mathcal{W}^{\text{local}}$); (g) final weights ($\mathcal{W}^{\text{final}}$), obtained via Eq. 1. Mean (red) and median (green) values are marked for reference.

## D    DISCUSSION

The implications of WAG extend beyond enhancing data-driven reasoning. By design, WAG enables users and prescribing healthcare professionals to contribute personalized information to the knowledge graph. For instance, a clinician may define individualized thresholds for blood pressure, while a user might incorporate contextual interpretations of stress levels based on lifestyle factors or recent events. This personalization enhances the system's adaptability to individual variations in health interpretation and reasoning. Moreover, by interacting with an LLM orchestrator, the personal knowledge graph can be continuously updated with minimal effort, reducing the burden of manual curation.

It is important to note that we did not manually verify the factual correctness of generated responses, as such evaluation would be prohibitively labor-intensive. The novelty of this work does not lie in improving factual accuracy, which could be enhanced by employing stronger models or complementary techniques, but rather in demonstrating how the integration of a wearable knowledge graph allows LLMs to deliver more insightful and contextually grounded findings.

Our human evaluation further underscores the challenges of this task. We observed relatively low inter-rater reliability (IRR) and modest correlation between LLM and human rankings, indicating that response evaluation is inherently subjective. These findings raise the possibility of employing LLMs themselves as cost-effective evaluators in settings where recruiting large numbers of medical experts is impractical. However, potential biases embedded in LLMs could be a critical limitation, requiring further investigation into when and how they can serve as fair substitutes for human evaluators. Especially, we found that in difficult cases, where responses are hard to differentiate across the defined dimensions, LLMs often assign identical ranks across all dimensions of a response. This observation led us to focus solely on overall quality in Experiments G and L. Future work should explore strategies to mitigate such biases while leveraging the scalability advantages of LLM-based evaluation.

## E  LLM USAGE

LLM serves as a core component of our approach, such as creating the query set and knowledge graph, etc. Additionally, it is applied to polish the phrasing and wording during the paper writeup.

# F ADDITIONAL RESULTS

Table 10: Main experiment - Comparison of our method with baselines across all query types.

| Query Type | Method | Rank | | | | | | | Overall |
|---|---|---|---|---|---|---|---|---|---|
| | | Insight | Relevance | Grounded | Personal | Clarity | Safety | Overall | Win Rate |
| General Knowledge | Base | 2.37 | 2.37 | 2.29 | 2.31 | 2.23 | 1.00 | 2.37 | 0.18 |
| | Rag | 2.09 | 2.09 | 2.10 | 2.12 | 1.98 | 1.00 | 2.09 | 0.21 |
| | WAG | **1.54** | **1.54** | **1.58** | **1.56** | **1.73** | 1.00 | **1.54** | **0.60** |
| Data Retrieval | Base | 2.68 | 2.66 | 2.43 | 2.63 | 2.35 | 1.00 | 2.68 | 0.08 |
| | Rag | 1.70 | 1.71 | 1.76 | 1.72 | **1.72** | 1.00 | 1.70 | 0.41 |
| | WAG | **1.62** | **1.63** | **1.67** | **1.64** | 1.77 | 1.00 | **1.62** | **0.51** |
| Trend Analysis | Base | 2.63 | 2.62 | 2.52 | 2.62 | 2.32 | 1.00 | 2.63 | 0.03 |
| | Rag | 2.02 | 2.03 | 2.07 | 2.03 | 1.90 | 1.00 | 2.02 | 0.22 |
| | WAG | **1.35** | **1.35** | **1.40** | **1.35** | **1.73** | 1.00 | **1.35** | **0.75** |
| Comparative Insight | Base | 2.67 | 2.66 | 2.44 | 2.65 | 2.24 | 1.00 | 2.67 | 0.04 |
| | Rag | 2.05 | 2.06 | 2.14 | 2.07 | 1.93 | 1.00 | 2.05 | 0.15 |
| | WAG | **1.28** | **1.28** | **1.40** | **1.28** | **1.76** | 1.00 | **1.28** | **0.80** |
| Anomaly Detection | Base | 2.70 | 2.70 | 2.51 | 2.68 | 2.22 | 1.00 | 2.70 | 0.01 |
| | Rag | 2.18 | 2.18 | 2.25 | 2.19 | 1.91 | 1.00 | 2.18 | 0.07 |
| | WAG | **1.12** | **1.12** | **1.21** | **1.12** | **1.78** | 1.00 | **1.12** | **0.92** |
| Actionable Advice | Base | 2.43 | 2.43 | 2.38 | 2.43 | 2.14 | 1.00 | 2.43 | 0.09 |
| | Rag | 2.09 | 2.09 | 2.10 | 2.09 | 1.95 | 1.00 | 2.09 | 0.21 |
| | WAG | **1.48** | **1.48** | **1.50** | **1.47** | **1.84** | 1.00 | **1.48** | **0.70** |
| Exploratory Analysis | Base | 2.63 | 2.63 | 2.53 | 2.63 | 2.21 | 1.00 | 2.63 | 0.03 |
| | Rag | 2.11 | 2.11 | 2.15 | 2.11 | 1.88 | 1.00 | 2.11 | 0.14 |
| | WAG | **1.26** | **1.26** | **1.32** | **1.26** | **1.85** | 1.00 | **1.26** | **0.83** |
| Metric Relationships | Base | 2.55 | 2.55 | 2.43 | 2.52 | 2.36 | 1.00 | 2.55 | 0.09 |
| | Rag | 1.92 | 1.92 | 1.98 | 1.94 | 1.91 | 1.00 | 1.92 | 0.30 |
| | WAG | **1.53** | **1.53** | **1.59** | **1.53** | **1.71** | 1.00 | **1.53** | **0.61** |
| Contextual Queries | Base | 2.50 | 2.50 | 2.41 | 2.49 | 2.32 | 1.00 | 2.50 | 0.09 |
| | Rag | 2.00 | 2.00 | 2.05 | 2.01 | 1.92 | 1.00 | 2.00 | 0.24 |
| | WAG | **1.49** | **1.50** | **1.53** | **1.50** | **1.75** | 1.00 | **1.49** | **0.67** |

Table 11: Main experiment - Comparison of our method with baselines across all time periods.

| Time Period | Method | Rank | | | | | | | Overall |
|---|---|---|---|---|---|---|---|---|---|
| | | Insight | Relevance | Grounded | Personal | Clarity | Safety | Overall | Win Rate |
| 1 | Base | 2.74 | 2.74 | 2.49 | 2.71 | 2.31 | 1.00 | 2.74 | 0.04 |
| | Rag | 1.91 | 1.91 | 1.98 | 1.92 | 1.80 | 1.00 | 1.91 | 0.24 |
| | WAG | **1.35** | **1.36** | **1.44** | **1.36** | **1.74** | 1.00 | **1.35** | **0.72** |
| 7 | Base | 2.65 | 2.63 | 2.49 | 2.62 | 2.26 | 1.00 | 2.65 | 0.04 |
| | Rag | 1.98 | 1.99 | 2.04 | 2.00 | 1.87 | 1.00 | 1.98 | 0.23 |
| | WAG | **1.37** | **1.38** | **1.41** | **1.37** | **1.79** | 1.00 | **1.37** | **0.74** |
| 14 | Base | 2.60 | 2.59 | 2.50 | 2.58 | 2.26 | 1.00 | 2.60 | 0.04 |
| | Rag | 2.00 | 2.00 | 2.04 | 2.01 | **1.84** | 1.00 | 2.00 | 0.24 |
| | WAG | **1.40** | **1.40** | **1.44** | **1.40** | 1.85 | 1.00 | **1.40** | **0.72** |
| 30 | Base | 2.50 | 2.50 | 2.42 | 2.49 | 2.23 | 1.00 | 2.50 | 0.08 |
| | Rag | 2.05 | 2.06 | 2.09 | 2.06 | 1.94 | 1.00 | 2.05 | 0.23 |
| | WAG | **1.44** | **1.45** | **1.49** | **1.45** | **1.80** | 1.00 | **1.44** | **0.69** |
| all | Base | 2.40 | 2.40 | 2.34 | 2.39 | 2.25 | 1.00 | 2.40 | 0.13 |
| | Rag | 2.03 | 2.03 | 2.04 | 2.03 | 1.94 | 1.00 | 2.03 | 0.27 |
| | WAG | **1.57** | **1.57** | **1.60** | **1.57** | **1.77** | 1.00 | **1.57** | **0.60** |

Table 12: Main experiment - Comparison of our method with baselines across all abnormal levels.

| Abnormal level | Method | Insight | Relevance | Grounded | Personal | Clarity | Safety | Overall | Win Rate |
|---|---|---|---|---|---|---|---|---|---|
| low | Base | 2.67 | 2.65 | 2.46 | 2.62 | 2.36 | 1.00 | 2.67 | 0.07 |
| | Rag | 1.82 | 1.83 | 1.87 | 1.84 | 1.77 | 1.00 | 1.82 | 0.33 |
| | WAG | **1.51** | **1.52** | **1.57** | **1.52** | **1.74** | 1.00 | **1.51** | **0.60** |
| medium | Base | 2.69 | 2.68 | 2.49 | 2.67 | 2.26 | 1.00 | 2.69 | 0.03 |
| | Rag | 2.06 | 2.07 | 2.14 | 2.08 | 1.90 | 1.00 | 2.06 | 0.15 |
| | WAG | **1.25** | **1.25** | **1.34** | **1.25** | **1.77** | 1.00 | **1.24** | **0.83** |
| high | Base | 2.62 | 2.62 | 2.51 | 2.62 | 2.19 | 1.00 | 2.62 | 0.03 |
| | Rag | 2.12 | 2.12 | 2.16 | 2.12 | 1.89 | 1.00 | 2.12 | 0.14 |
| | WAG | **1.26** | **1.26** | **1.32** | **1.26** | **1.85** | 1.00 | **1.26** | **0.83** |
| other | Base | 2.50 | 2.49 | 2.38 | 2.47 | 2.27 | 1.00 | 2.50 | 0.11 |
| | Rag | 1.92 | 1.92 | 1.95 | 1.93 | 1.87 | 1.00 | 1.92 | 0.32 |
| | WAG | **1.58** | **1.59** | **1.62** | **1.59** | **1.78** | 1.00 | **1.58** | **0.58** |

Table 13: Experiment-G - Comparison of different weighting within global modeling across datasets.

| Dataset | $\mathcal{W}^{global}$ | | $\mathcal{W}^{ind}$ | | $\mathcal{W}^{pop}$ | | $\mathcal{W}^{prior}$ | |
|---|---|---|---|---|---|---|---|---|
| | Mean | Win Rate | Mean | Win Rate | Mean | Win Rate | Mean | Win Rate |
| Globem | **2.14** | 0.30 | 2.62 | 0.17 | 2.50 | 0.22 | 2.22 | **0.36** |
| IFH Affect | **2.08** | **0.34** | 2.42 | 0.23 | 2.40 | 0.25 | 2.39 | 0.27 |
| Lifesnap | **2.14** | 0.28 | 2.41 | 0.24 | 2.32 | **0.30** | 2.34 | 0.28 |
| Pmdata | **2.28** | 0.30 | 2.42 | 0.23 | 2.40 | 0.23 | 2.32 | **0.32** |
| Average | **2.16** | **0.31** | 2.47 | 0.22 | 2.40 | 0.25 | 2.32 | **0.31** |

Table 14: Experiment-G - Comparison of different weighting within global modeling across query types.

| Dataset | $\mathcal{W}^{global}$ | | $\mathcal{W}^{ind}$ | | $\mathcal{W}^{pop}$ | | $\mathcal{W}^{prior}$ | |
|---|---|---|---|---|---|---|---|---|
| | Mean | Win Rate | Mean | Win Rate | Mean | Win Rate | Mean | Win Rate |
| General Knowledge | **2.00** | **0.50** | 2.50 | 0.00 | 3.00 | 0.00 | **2.00** | **0.50** |
| Data Retrieval | **2.23** | 0.31 | 2.38 | **0.38** | 2.54 | 0.15 | **2.23** | 0.23 |
| Trend Analysis | **2.10** | **0.34** | 2.53 | 0.19 | 2.58 | 0.17 | 2.33 | 0.33 |
| Comparative Insight | **2.14** | **0.34** | 2.47 | 0.22 | 2.50 | 0.23 | 2.34 | 0.29 |
| Anomaly Detection | 2.26 | 0.25 | 2.47 | 0.21 | **2.21** | **0.32** | 2.48 | 0.29 |
| Actionable Advice | **2.06** | **0.37** | 2.43 | 0.26 | 2.26 | 0.28 | 2.38 | 0.28 |
| Exploratory Analysis | **2.16** | 0.30 | 2.47 | 0.22 | 2.44 | 0.25 | 2.21 | **0.33** |
| Average | **2.14** | **0.34** | 2.46 | 0.21 | 2.50 | 0.20 | 2.28 | 0.32 |

Table 15: Experiment-G - Comparison of different weighting within global modeling across time periods.

| Dataset | $\mathcal{W}^{global}$ | | $\mathcal{W}^{ind}$ | | $\mathcal{W}^{pop}$ | | $\mathcal{W}^{prior}$ | |
|---|---|---|---|---|---|---|---|---|
| | Mean | Win Rate | Mean | Win Rate | Mean | Win Rate | Mean | Win Rate |
| 1 | **2.15** | **0.31** | 2.49 | 0.21 | 2.35 | 0.28 | 2.29 | 0.31 |
| 7 | **2.20** | **0.29** | 2.40 | 0.24 | 2.25 | 0.28 | 2.42 | 0.28 |
| 14 | **2.08** | 0.29 | 2.52 | 0.21 | 2.44 | 0.24 | 2.29 | **0.31** |
| 30 | **2.18** | **0.32** | 2.45 | 0.22 | 2.50 | 0.21 | 2.31 | 0.32 |
| Average | **2.16** | **0.30** | 2.47 | 0.22 | 2.39 | 0.26 | 2.33 | **0.30** |

Table 16: Experiment-G - Comparison of different weighting within global modeling across abnormal levels.

| Dataset | $\mathcal{W}^{global}$ | | $\mathcal{W}^{ind}$ | | $\mathcal{W}^{pop}$ | | $\mathcal{W}^{prior}$ | |
|---|---|---|---|---|---|---|---|---|
| | Mean | Win Rate | Mean | Win Rate | Mean | Win Rate | Mean | Win Rate |
| low | **2.03** | **0.38** | 2.57 | 0.21 | 2.54 | 0.16 | 2.31 | 0.31 |
| medium | **2.19** | **0.31** | 2.47 | 0.20 | 2.43 | 0.25 | 2.35 | **0.31** |
| high | **2.16** | 0.30 | 2.45 | 0.24 | 2.36 | 0.26 | 2.29 | **0.31** |
| Average | **2.12** | **0.33** | 2.50 | 0.22 | 2.45 | 0.23 | 2.32 | 0.31 |

Table 17: Experiment-L - Evaluation of the effectiveness of local modeling across datasets

| Dataset | $\mathcal{W}^{final}$ | | $\mathcal{W}^{global}$ | | $\mathcal{W}^{local}$ | |
|---|---|---|---|---|---|---|
| | Mean | Win Rate | Mean | Win Rate | Mean | Win Rate |
| Globem | **1.90** | **0.36** | 1.98 | 0.33 | 2.01 | 0.35 |
| IFH Affect | **1.88** | **0.38** | 1.98 | 0.33 | 2.02 | 0.34 |
| Lifesnap | **1.85** | **0.38** | 1.94 | 0.37 | 2.10 | 0.29 |
| Pmdata | **1.89** | **0.36** | 2.01 | **0.36** | 1.99 | 0.33 |
| Average | **1.88** | **0.37** | 1.98 | 0.35 | 2.03 | 0.33 |

Table 18: Experiment-L - Evaluation of the effectiveness of local modeling across query types.

| Query Type | $\mathcal{W}^{final}$ | | $\mathcal{W}^{global}$ | | $\mathcal{W}^{local}$ | |
|---|---|---|---|---|---|---|
| | Mean | Win Rate | Mean | Win Rate | Mean | Win Rate |
| General Knowledge | **1.48** | **0.63** | 2.04 | 0.33 | 2.30 | 0.11 |
| Data Retrieval | 1.90 | 0.31 | **1.75** | **0.44** | 2.23 | 0.25 |
| Trend Analysis | 1.94 | **0.40** | **1.88** | 0.34 | 2.06 | 0.32 |
| Comparative Insight | 1.95 | 0.34 | **1.91** | **0.40** | 2.09 | 0.28 |
| Anomaly Detection | **1.91** | 0.34 | 2.02 | 0.32 | 1.93 | **0.40** |
| Actionable Advice | **1.90** | 0.37 | 1.96 | **0.37** | 1.96 | 0.33 |
| Exploratory Analysis | **1.84** | **0.39** | 2.02 | 0.32 | 2.04 | 0.33 |
| Average | **1.85** | **0.40** | 1.94 | 0.36 | 2.09 | 0.29 |

Table 19: Experiment-L - Evaluation of the effectiveness of local modeling across time periods.

| Time Period | $\mathcal{W}^{final}$ | | $\mathcal{W}^{global}$ | | $\mathcal{W}^{local}$ | |
|---|---|---|---|---|---|---|
| | Mean | Win Rate | Mean | Win Rate | Mean | Win Rate |
| 1 | **1.89** | **0.36** | 1.99 | 0.33 | 1.99 | **0.36** |
| 7 | 1.91 | 0.35 | **1.89** | **0.40** | 2.08 | 0.30 |
| 14 | **1.85** | 0.36 | 1.96 | **0.37** | 2.11 | 0.30 |
| 30 | **1.87** | **0.40** | 2.02 | 0.33 | 2.01 | 0.32 |
| Average | **1.88** | **0.37** | 1.96 | 0.36 | 2.05 | 0.32 |

Table 20: Experiment-L - Evaluation of the effectiveness of local modeling across abnormal levels.

| Abnormal Level | $\mathcal{W}^{final}$ | | $\mathcal{W}^{global}$ | | $\mathcal{W}^{local}$ | |
|---|---|---|---|---|---|---|
| | Mean | Win Rate | Mean | Win Rate | Mean | Win Rate |
| low | **1.74** | **0.45** | 1.92 | 0.36 | 2.20 | 0.22 |
| medium | **1.90** | **0.36** | 1.95 | **0.36** | 2.04 | 0.32 |
| high | **1.89** | **0.36** | 2.01 | 0.33 | 1.99 | 0.35 |
| Average | **1.84** | **0.39** | 1.96 | 0.35 | 2.08 | 0.30 |

(a) IRR and Spearman correlation between LLM and human responses (IRR computed via Krippendorff's alpha).

| Experiment | Main | 1 | 2 |
|---|---|---|---|
| **IRR** | 0.38 | 0.26 | 0.32 |
| **Correlation** | 0.55 | 0.14 | 0.13 |

(b) Human Evaluator Results for Main Experiment

| Evaluator | WAG | Rag | Base |
|---|---|---|---|
| E1 | 1.37 | 2.01 | 2.60 |
| E2 | 1.49 | 1.90 | 2.33 |
| E3 | 1.56 | 1.80 | 2.41 |
| **Human Avg** | 1.47 | 1.90 | 2.45 |
| **LLM** | 1.41 | 1.98 | 2.61 |

Table 22: Human Evaluator Results for Experiment-G.

| Evaluator | $\mathcal{W}^{global}$ | $\mathcal{W}^{ind}$ | $\mathcal{W}^{pop}$ | $\mathcal{W}^{prior}$ |
|---|---|---|---|---|
| E1 | 2.31 | 2.57 | 2.55 | 2.42 |
| E2 | 2.35 | 2.40 | 2.32 | 2.58 |
| E3 | 2.26 | 2.56 | 2.52 | 2.29 |
| **Human Avg** | 2.31 | 2.51 | 2.46 | 2.43 |
| **LLM** | 2.16 | 2.47 | 2.40 | 2.32 |

Table 23: Human Evaluator Results for Experiment-L.

| Evaluator | $\mathcal{W}^{final}$ | $\mathcal{W}^{global}$ | $\mathcal{W}^{local}$ |
|---|---|---|---|
| E1 | 1.87 | 1.91 | 2.19 |
| E2 | 1.83 | 2.05 | 2.04 |
| E3 | 2.07 | 1.90 | 1.91 |
| **Human Avg** | 1.92 | 1.95 | 2.05 |
| **LLM** | 1.88 | 1.98 | 2.03 |

# G  DETAILED METHOD

## G.1  GLOBAL MODELING-HBM DERIVATION

This provides a comprehensive derivation of the posterior distribution for the latent long-term edge weight vector $\Theta_x^s$ for a subject $s$ and source node $x$. The model integrates a prior distribution, population-level data, and individual-level data within a conjugate Gaussian framework.

$$\Theta_x^s = [\theta_{x,y_1}^s, \ldots, \theta_{x,y_{|Y|}}^s]^\top \in \mathbb{R}^{|Y|}$$

MODEL SPECIFICATION

$$\Theta_x^s \sim \mathcal{N}(\mu_x^{\text{prior}}, \Sigma_x^{\text{prior}}), \tag{2}$$
$$R_x^{\text{pop}} \mid \Theta_x^s \sim \mathcal{N}(\Theta_x^s, V_x^{\text{pop}}), \tag{3}$$
$$R_x^s \mid \Theta_x^s \sim \mathcal{N}(\Theta_x^s, V_x^s), \tag{4}$$

where $R_x^{\text{pop}}, R_x^s, \mu_x^{\text{prior}} \in \mathbb{R}^{|Y|}$ and $\Sigma_x^{\text{prior}}, V_x^{\text{pop}}, V_x^s$ are covariance matrices.

By Bayes' rule and conditional independence of the two observation sources given $\Theta_x^s$,

$$p(\Theta_x^s \mid R_x^{\text{pop}}, R_x^s) \propto p(\Theta_x^s)\, p(R_x^{\text{pop}} \mid \Theta_x^s)\, p(R_x^s \mid \Theta_x^s).$$

Equivalently, this can be viewed as updating the population-informed posterior with the individual's data:

$$p(\Theta_x^s \mid R_x^{\text{pop}}, R_x^s) \propto p(R_x^s \mid \Theta_x^s)\, p(\Theta_x^s \mid R_x^{\text{pop}}).$$

The (negative) log of the posterior as follows:

$$\mathcal{L}(\Theta) \equiv -\log p(\Theta_x^s \mid R_x^{\text{pop}}, R_x^s) + C$$

$$= \tfrac{1}{2}(\Theta - \mu^{\text{prior}})^\top (\Sigma^{\text{prior}})^{-1}(\Theta - \mu^{\text{prior}})$$

$$+ \tfrac{1}{2}(R^{\text{pop}} - \Theta)^\top (V^{\text{pop}})^{-1}(R^{\text{pop}} - \Theta)$$

$$+ \tfrac{1}{2}(R^s - \Theta)^\top (V^s)^{-1}(R^s - \Theta),$$

To simplify notation for the derivation, subscript $x$ and superscript $s$ inside this derivation are dropped: $\Theta \equiv \Theta_x^s$, $\mu^{\text{prior}} \equiv \mu_x^{\text{prior}}$, $R^{\text{pop}} \equiv R_x^{\text{pop}}$, $R^s \equiv R_x^s$, etc.

Expand each quadratic term and collect terms in $\Theta$:

$$\mathcal{L}(\Theta) = \tfrac{1}{2}\Theta^\top (\Sigma^{\text{prior}})^{-1}\Theta - (\Sigma^{\text{prior}})^{-1}\mu^{\text{prior}\top}\Theta + \tfrac{1}{2}\mu^{\text{prior}\top}(\Sigma^{\text{prior}})^{-1}\mu^{\text{prior}}$$

$$+ \tfrac{1}{2}\Theta^\top (V^{\text{pop}})^{-1}\Theta - (V^{\text{pop}})^{-1}R^{\text{pop}\top}\Theta + \tfrac{1}{2}R^{\text{pop}\top}(V^{\text{pop}})^{-1}R^{\text{pop}}$$

$$+ \tfrac{1}{2}\Theta^\top (V^s)^{-1}\Theta - (V^s)^{-1}R^{s\top}\Theta + \tfrac{1}{2}R^{s\top}(V^s)^{-1}R^s.$$

Collecting the quadratic (in $\Theta$) and linear terms yields

$$\mathcal{L}(\Theta) = \tfrac{1}{2}\Theta^\top \Lambda \Theta - b^\top \Theta + \text{const},$$

where

$$\Lambda \equiv (\Sigma^{\text{prior}})^{-1} + (V^{\text{pop}})^{-1} + (V^s)^{-1}, \tag{5}$$

$$b \equiv (\Sigma^{\text{prior}})^{-1}\mu^{\text{prior}} + (V^{\text{pop}})^{-1}R^{\text{pop}} + (V^s)^{-1}R^s. \tag{6}$$

Complete the square for the quadratic form:

$$\mathcal{L}(\Theta) = \tfrac{1}{2}(\Theta^\top \Lambda \Theta - 2b^\top \Theta) + \text{const}$$

$$= \tfrac{1}{2}(\Theta - \Lambda^{-1}b)^\top \Lambda (\Theta - \Lambda^{-1}b) - \tfrac{1}{2}b^\top \Lambda^{-1}b + \text{const}.$$

Therefore the posterior is Gaussian,

$$\Theta_x^s \mid R_x^{\text{pop}}, R_x^s \sim \mathcal{N}(\mu, \Sigma),$$

with

$$\Sigma = \Lambda^{-1} = \left((\Sigma^{\text{prior}})^{-1} + (V^{\text{pop}})^{-1} + (V^s)^{-1}\right)^{-1}, \tag{7}$$

$$\mu = \Sigma b = \Sigma\left((\Sigma^{\text{prior}})^{-1}\mu^{\text{prior}} + (V^{\text{pop}})^{-1}R^{\text{pop}} + (V^s)^{-1}R^s\right). \tag{8}$$

**Note**: Our method is not a typical fully generative hierarchical Bayesian model. In a standard formulation, the population distribution is treated as a set of latent hyperparameters with their own priors, and inference is carried out via MCMC or variational methods. While this approach is flexible, it typically requires computationally intensive sampling, which must be repeated at inference time. We believe such sampling is not appropriate for our setting, where repeated, efficient inference is required. Instead, our approach adopts a simplified empirical Bayes formulation with closed-form Gaussian updates. This allows us to retain the population-to-individual hierarchy while ensuring computational tractability.

### G.2 LOCAL MODELING

weights capture context-sensitive relationships over the past $k^q$ days relative to query time $t^q$. For node $x$ and neighbor $y$, the normalized abnormality score is:

$$\zeta_y = \frac{1}{k^q} \sum_{i=0}^{k^q-1} \left| \frac{v_{y,t^q-i} - \mu_y}{\sigma_y} \right|,$$

where $\mu_y$ and $\sigma_y$ are the historical mean and standard deviation.

We define the short-term weight by convexly mixing $\zeta_y$ and its complement:

$$w_{x,y}^{\text{short}} = \eta^q \zeta_y + (1 - \eta^q)(1 - \zeta_y).$$

Expanding gives:

$$w_{x,y}^{\text{short}} = (2\eta^q - 1)\zeta_y + (1 - \eta^q),$$

which is bounded in $[0, 1]$ for $\eta^q, \zeta_y \in [0, 1]$.

# H NOTATION

Table 24: Question Categories for Health Data Analysis

| Category | Openness | Description | Example Questions |
|---|---|---|---|
| **Single-Entity** | | | |
| General Knowledge | 0.2-0.4 | Definition and basic understanding of metrics, optimal ranges, and benchmarks | What is HRV and its optimal range?
What is resting heart rate? |
| Data Retrieval | 0.1-0.3 | Specific time-bound numerical queries | What was my step count this week?
Average deep sleep minutes past 14 days? |
| Trend Analysis | 0.4-0.6 | Trend identification and behavioral patterns | Identify trends in my daily movement last 30 days.
When do I typically have my most active days? |
| Comparative Insight | 0.5-0.7 | Time-based comparisons between periods | How does this week's activity compare to last week?
Has my sleep duration improved this month? |
| Anomaly Detection | 0.6-0.8 | Outlier identification and unusual deviations | Any unusual sleep metrics this month?
Was there an abnormal recovery time this week? |
| Actionable Advice | 0.3-0.5 | Actionable suggestions based on current data | How to improve my sleep quality?
Ways to increase my activity score? |
| Exploratory Analysis | 0.7-1.0 | Multi-factor investigations | Why am I tired despite sleeping 8 hours?
Do you think I'm stressed recently? |
| **Multi Entity Category** | | | |
| Metric Relationships | 0.4-0.6 | Exploration of correlations or interactions between two or more health metrics over a period of time | Did my activity levels impact my readiness score?
How does my REM sleep duration correlate with my stress levels for the past 30 days?
What's the relationship between my exercise intensity and recovery time? |
| Contextual Queries | 0.5-0.7 | Questions that examine relationships between a health metric and contextual factors (e.g., stress, sleep, activity) | Do my sleep disturbances increase on days with higher stress scores last week?
Is there a pattern in my heart rate variability on days I have a higher activity level for the past month? |

Table 25: Notations and Descriptions

| Notation | Description |
|---|---|
| $D$ | a dataset |
| $s \in \mathcal{S}_{sel}$ | participants selected |
| $m \in \mathcal{M}$ | a concept of health metric, eg. heart rate, mood |
| $t \in \mathcal{T}$ | all timestamps (day) of data |
| $v_t$ | data value at timestamp(day) $t$ |
| $\mathrm{MD}_s$ | missing data percentage for a subject $s$'s data |
| $\mathrm{VL}_s$ | valid period length for a subject $s$'s data |
| $\mathrm{CV}_s$ | coefficient of variation for a subject $s$'s data. |
| $\mathrm{MI}_p$ | pairwise mutual information for a subject $s$'s data |
| $\zeta_{k,t} \in Z$ | abnormal level of data for the past $k$ days before $t$ days |
| $k \in \mathcal{K}$ | window size |
| $\mathcal{I}$ | input tuple to generate query |
| $q \in \mathcal{Q}$ | query set |
| $\eta$ | openness score |
| $c \in \mathcal{C}$ | a category of health metric,eg. sleep, activity |
| $\mathcal{G} = (\mathcal{V}, \mathcal{E})$ | knowledge graph |
| $v \in \mathcal{V}$ | node |
| $e_{i,j} \in \mathcal{E}$ | the edge encodes the relationship between nodes $v_i$ and $v_j$ |
| $w_{i,j} \in \mathcal{W}$ | weight of the edge $e_{i,j}$ |
| $\theta_x^s$ | the latent vector of edge weight $\mathcal{W}$ |
| $\mu^{\mathrm{prior}}, \Sigma^{\mathrm{prior}}$ | prior distribution of $\theta^s$ |
| $R^{\mathrm{pop}}$ | spearman correlations between historical data across the dataset. |
| $R^s$ | historical data of correlations computed from user $s$'s data. |
| $\mu^{\mathrm{pop}}, \Sigma^{\mathrm{pop}}$ | distribution of $\theta_x^s$ after posterior update of $R_x^{\mathrm{pop}}$ |
| $\mu_x^s(\mathcal{W}^{global}), \Sigma_x^s$ | distribution of $\theta_x^s$ after further posterior update of $R_x^{\mathrm{ind}}$ |
| $\mathcal{W}^{local}$ | weights obtained from local modeling |
| $\beta$ | hyperparameter to control $\mathcal{W}_{global}$ and $\mathcal{W}_{local}$ |
| $\delta$ | match threshold for similarity match $sim(v_i, v_j)$ |
| $\kappa$ | max number of related nodes that will be retrieved |
| $\alpha_{\mathrm{pop}}, \alpha_{\mathrm{ind}}$ | hyperparameters to control the role of $R_x^{\mathrm{pop}}$ and $R_x^{\mathrm{ind}}$ respectively in HBM modeling |

Table 26: Node Field Descriptions

| Field | Description |
|---|---|
| ID | Id of the node |
| Name | Name of the node |
| Description | Description of the node |
| Range | Range of values with units |
| Recommendation | Recommendation for improvement |
| DataSource | Data source specification including dataset name, feature name, description, range, unit, and type, path to data |
| Weight | Importance weight for sorting (higher = more important) |
| If_data_associated | Flag indicating data association status |
| Name_embedding | Name-based embedding vector |
| Semantic_embedding | Textual embedding vector |
| Graph_embedding | Graph structure embedding |
| Umls_name | Standardized UMLS name |
| CUI | UMLS Concept Unique Identifier |
| Umls_definition | Formal UMLS concept definition |
| Raw_web_result | Unprocessed web extraction data |

Table 27: Edge Field Descriptions

| Field | Description |
|---|---|
| ID | Unique relationship identifier |
| Node_1_name | Name of first node in relationship |
| Node_1_description | Description of first node |
| Node_1_id | Unique identifier of first node |
| Node_2_name | Name of second node in relationship |
| Node_2_description | Description of second node |
| Node_2_id | Unique identifier of second node |
| weight | Edge weight (default is generated from LLM) |
| description | Textual description of relationship nature/purpose |
| description_embedding | Semantic embedding vector for relationship description |
| raw_web_result | Reference to raw web search results |

Table 28: Initial Health Metrics

| Category | Metrics |
|---|---|
| **Physiological** | Heart rate, Heart rate variability, Blood pressure, Blood oxygen saturation, Pulse wave velocity, Cardiac output, Peripheral blood flow, Respiratory rate, Oxygen uptake, Carbon dioxide exhalation rate, Lung volume, Breathing rhythm, Muscle activity, Blood glucose levels, Lactate levels, Basal metabolic rate, Core body temperature, Brain activity, Skin temperature, Sweat rate, Electrolyte concentration, Skin hydration levels, Skin conductance, pH of sweat |
| **Sleep** | Sleep stages (light, deep, REM), Sleep apnea events, Total sleep duration, Sleep onset latency, Wake after sleep onset, Sleep efficiency, Arousals per night, Breathing irregularities, Snoring patterns, Circadian rhythm patterns |
| **Activity** | Steps taken, Distance traveled, Active minutes, Energy expenditure, Activity level, Running dynamics, Balance and stability, Joint movement and flexibility |
| **Mental** | Mental stress, Mental fatigue, Mental workload, Engagement, Cognitive load, Memory performance, Decision-making speed, Mood |
| **Environmental** | Ambient temperature, Humidity, Barometric pressure, UV radiation exposure, Air quality, Noise levels, Electromagnetic field exposure, Altitude |
| **Lifestyle** | GPS location, Travel patterns, Proximity to other devices or people, Interaction frequency with social contacts, Time spent on specific activities, Screen interaction patterns, Daily routine adherence |
| **Demographic** | Gender, Age, personality |

Table 29: Metrics available across datasets. ✓ indicates presence, ✗ absence.

| Type | Metric | Globem | Ifh_affect | Lifesnap | Pmdata |
|---|---|---|---|---|---|
| **Physiological** | Blood oxygen saturation | ✗ | ✗ | ✓ | ✗ |
| | Skin conductance | ✗ | ✗ | ✓ | ✗ |
| | HRV RMSSD | ✗ | ✓ | ✓ | ✗ |
| | Resting heart rate | ✗ | ✓ | ✓ | ✗ |
| | Nightly skin temperature | ✗ | ✗ | ✓ | ✗ |
| | Sleep breathing rate | ✗ | ✓ | ✓ | ✗ |
| | Temperature Variation | ✗ | ✓ | ✓ | ✗ |
| | VO2 Max | ✗ | ✗ | ✓ | ✗ |
| **Sleep** | Total sleep duration | ✓ | ✓ | ✓ | ✓ |
| | Sleep onset latency | ✓ | ✓ | ✓ | ✓ |
| | Wake after sleep onset | ✓ | ✓ | ✓ | ✓ |
| | Sleep efficiency | ✓ | ✓ | ✓ | ✓ |
| | Circadian rhythm patterns | ✓ | ✗ | ✗ | ✗ |
| | Asleep duration | ✓ | ✓ | ✓ | ✓ |
| | Light sleep duration | ✗ | ✓ | ✓ | ✓ |
| | Deep sleep duration | ✗ | ✓ | ✓ | ✓ |
| | Rem sleep duration | ✗ | ✓ | ✓ | ✓ |
| | Bedtime start time | ✗ | ✓ | ✓ | ✓ |
| | Bedtime end time | ✗ | ✓ | ✓ | ✓ |
| | Midpoint of sleep | ✗ | ✓ | ✗ | ✗ |
| | Time after wakeup | ✓ | ✗ | ✓ | ✓ |
| **Activity** | Steps taken | ✓ | ✓ | ✓ | ✓ |
| | Distance traveled | ✗ | ✓ | ✓ | ✓ |
| | Active time | ✓ | ✓ | ✓ | ✓ |
| | Energy expenditure | ✗ | ✓ | ✓ | ✓ |
| | Resting time | ✗ | ✓ | ✗ | ✗ |
| | Inactive time | ✗ | ✗ | ✓ | ✓ |
| | Lightly active time | ✗ | ✓ | ✓ | ✓ |
| | Moderately active time | ✗ | ✓ | ✓ | ✓ |
| | Highly active time | ✗ | ✓ | ✓ | ✓ |
| | Exercise | ✗ | ✗ | ✓ | ✓ |
| | Step goal | ✗ | ✗ | ✓ | ✗ |
| **Mental** | Mental stress | ✓ | ✗ | ✗ | ✓ |
| | Mental fatigue | ✗ | ✗ | ✗ | ✓ |
| | Mood | ✗ | ✗ | ✓ | ✓ |
| | PANAS positive affect | ✓ | ✓ | ✓ | ✗ |
| | PANAS negative affect | ✓ | ✓ | ✓ | ✗ |
| | Anxiety | ✓ | ✗ | ✗ | ✗ |
| | Depression | ✓ | ✗ | ✗ | ✗ |
| **Environmental** | Barometric pressure | ✗ | ✓ | ✗ | ✗ |
| **Lifestyle** | Lifelog | ✗ | ✗ | ✓ | ✓ |
| | Number of calls | ✓ | ✗ | ✗ | ✗ |
| | Entropy of call duration | ✓ | ✗ | ✗ | ✗ |
| | Number of phone unlock | ✓ | ✗ | ✗ | ✗ |
| | Duration of phone unlock | ✓ | ✗ | ✗ | ✗ |
| | Time at home | ✓ | ✗ | ✗ | ✗ |
| | The Radius of Gyration | ✓ | ✗ | ✗ | ✗ |
| | Maximum distance from home | ✓ | ✗ | ✗ | ✗ |
| **Demographic** | Age | ✗ | ✗ | ✓ | ✗ |
| | Gender | ✗ | ✗ | ✓ | ✗ |
| | Bmi | ✗ | ✗ | ✓ | ✗ |
| | Personality | ✓ | ✗ | ✓ | ✗ |

# I  EVAL-UI

**Human Evaluation Instructions: Comparing Responses to Health Questions**

You are asked to evaluate and rank the quality of responses generated by multiple methods for a user's health-related question. Each method may use contextual information such as wearable data, time-series patterns, or other relevant health knowledge.

Evaluate responses based on the following dimensions:

- **Insightfulness (most important)**: Does the response provide meaningful, actionable insights beyond the obvious?
- **Relevance**: Is the response relevant and does it provide novel information?
- **Groundedness**: Are factual claims supported by the provided content or trusted sources?
- **Personalization**: Does the response meaningfully incorporate the user's context?
- **Clarity**: Is the response clearly written and easy to understand?
- **Absence of harmful content**: Is the response free from misleading, unsafe, or inappropriate information?

**Ranking Rules:** Assign rank 1 to the best response(s). Multiple responses can share the same rank if they are indistinguishable. Lower numbers indicate better performance.

You will see a user's question followed by responses from different methods. Compare them carefully and assign ranks based on overall quality.

**Load Data from File**

Choose File  ui_dict_sam..._global.json  Load File
Loaded ui_dict_sampled_global.json successfully.

Back  Index: 4 / 137  Enter key to jump  Go  Next  Download All Ranks

**Question:** What lifestyle changes could improve my sleep efficiency which has been unusually low over the past 30 days?
**Q_ID:** 06ee73f7-508c-4a8d-a9dd-637c6afbf7a2

Rank this result:  1  2  3  4

**Response**

Based on your sleep efficiency data and related metrics over the past 30 days, here are evidence-based lifestyle recommendations to improve your sleep quality:

1. **Sleep Schedule Optimization**
- Your data shows excellent sleep efficiency (consistently 90%+), suggesting your current routine is working well. Maintain your consistent bedtime/waketime schedule, as this is the strongest protective factor evident in your data.

2. **Activity Timing Adjustment**
- While your active minutes (avg ~300/day) exceed recommendations, note that January 29 showed both low activity (163min) and slightly reduced sleep efficiency (98% vs typical 99%). Consider:

**Primary Metrics**

**Sleep efficiency:**

**Related Metrics**

**Active time:**

**Context**

Matched nodes:

Sleep efficiency:
description: Sleep Efficiency (SE) is the percentage of time spent asleep compared to the total time in bed, calculated as the ratio of Total Sleep Time (TST) to Time in Bed (TIB) multiplied by 100. It is a critical metric in assessing sleep quality and identifying disorders like insomnia, with higher values indicating more efficient sleep. Wearable sensors estimate SE using accelerometers or heart rate variability to determine sleep and wake states. range: Normal SE is 80%-90% for healthy adults, with values below 80% indicating potential sleep disturbances. Units: percentage (%). recommendation: To improve sleep efficiency, maintain a consistent sleep schedule, avoid screen

---

Rank this result:  1  2  3  4

**Response**

- Address wakefulness: Your WASO averages 45 minutes (above the 30-minute healthy threshold). Implement:
- Consistent bedtime/wake time (even weekends)
- 1-hour pre-sleep relaxation routine
- Cool (18-20°C), dark sleeping environment

3. **Behavioral Timing**
- Avoid exercise within 4 hours of bedtime
- Limit caffeine after 2pm and alcohol within 3 hours of sleep
- Reduce screen exposure 1 hour before bed (blue light disrupts melatonin)

4. **Pattern Recognition**
- Your best efficiency (99%) occurred on days with:
- Light activity >200min (Jan

**Related Metrics**

**Inactive time:**

**Context**

Matched nodes:

Sleep efficiency:
description: Sleep Efficiency (SE) is the percentage of time spent asleep compared to the total time in bed, calculated as the ratio of Total Sleep Time (TST) to Time in Bed (TIB) multiplied by 100. It is a critical metric in assessing sleep quality and identifying disorders like insomnia, with higher values indicating more efficient sleep. Wearable sensors estimate SE using accelerometers or heart rate variability to determine sleep and wake states. range: Normal SE is 80%-90% for healthy adults, with values below 80% indicating potential sleep disturbances. Units: percentage (%). recommendation: To improve sleep efficiency, maintain a consistent sleep schedule, avoid screen

---

Rank this result:  1  2  3  4

**Response**

Based on your sleep efficiency data and related metrics, here are evidence-based lifestyle recommendations to improve your sleep quality:

1. **Sleep Schedule Optimization**:
- Your data shows excellent sleep efficiency (consistently 90%+), suggesting your current bedtime/waketime schedule, as this reinforces circadian rhythms.

2. **Activity Modifications**:
- Your active minutes (avg ~300/day) far exceed recommendations. Consider shifting some activity to earlier in the day - high energy expenditure late in the evening (like your 4399 kcal day on 1/30) may cause sleep fragmentation

**Related Metrics**

**Active time:**

**Context**

Matched nodes:

Sleep efficiency:
description: Sleep Efficiency (SE) is the percentage of time spent asleep compared to the total time in bed, calculated as the ratio of Total Sleep Time (TST) to Time in Bed (TIB) multiplied by 100. It is a critical metric in assessing sleep quality and identifying disorders like insomnia, with higher values indicating more efficient sleep. Wearable sensors estimate SE using accelerometers or heart rate variability to determine sleep and wake states. range: Normal SE is 80%-90% for healthy adults, with values below 80% indicating potential sleep disturbances. Units: percentage (%). recommendation: To improve sleep efficiency, maintain a consistent sleep schedule, avoid screen

---

Rank this result:  1  2  3  4

**Response**

Based on your sleep efficiency data and related metrics over the past 30 days, here are evidence-based lifestyle recommendations to improve your sleep quality:

1. **Sleep Schedule Optimization**
- Your sleep duration shows significant variability (ranging from 5 to 8.5 hours). Implement a consistent bedtime and wake-up time (±30 minutes) to regulate your circadian rhythm.

2. **Wakefulness Reduction**
- Your Wake After Sleep Onset (WASO) frequently exceeds the 30-minute threshold (peaking at 62 minutes).
- Action: Create a sleep-conducive environment by:
• Maintaining bedroom temperature at 18-20°C

**Primary Metrics**

**Sleep efficiency:**

**Related Metrics**

**Asleep duration:**

**Context**

Matched nodes:

Sleep efficiency:
description: Sleep Efficiency (SE) is the percentage of time spent asleep compared to the total time in bed, calculated as the ratio of Total Sleep Time (TST) to Time in Bed (TIB) multiplied by 100. It is a critical metric in assessing sleep quality and identifying disorders like insomnia, with higher values indicating more efficient sleep. Wearable sensors estimate SE using accelerometers or heart rate variability to determine sleep and wake states. range: Normal SE is 80%-90% for healthy adults, with values below 80% indicating potential sleep disturbances. Units: percentage (%). recommendation: To improve sleep efficiency, maintain a consistent sleep schedule, avoid screen

## J  QUALITATIVE EXAMPLES

Note: Some lengthy content has been truncated. Primary nodes are highlighted in blue, related nodes in orange, and key findings in yellow.

**Example 1:** A simple data retrieval question, where the incorporation of the KG into the query provides only marginal benefit.
**Query:** What types of exercise did I engage in most frequently over the past 7 days?
*Openness - 0.3 Query type - Data Retrieval*

| Base(Rank-2) – Context | Rag(Rank-1) – Context | WAG(Rank-3) – Context |
|---|---|---|
| Matched nodes:

Exercise :
sensor specific information if available:
1.description: daily log of exercise in detail
2.range:
3.unit:
Data:
— date — Exercise —
— 2019-11-15 — [{'logId': 26742666940, 'activityName': 'Walk', 'activityTypeId': 90013, 'activityLevel': [{'minutes': 0, 'name': 'sedentary'}, {'minutes': 0, 'name': 'lightly'} ... [truncated] | Matched nodes:

Exercise :
description: Exercise refers to planned, structured, and repetitive physical activity aimed at improving or maintaining physical fitness and overall health. It includes activities such as aerobic exercise, strength training, and flexibility exercises, which enhance cardiovascular health, muscle strength, and mental well-being. Exercise is a key component of a healthy lifestyle ... [truncated] | Matched nodes:

Exercise :
description: Exercise refers to planned ... [truncated] Recent 7-day value deviates from the individual's average by nan standard deviations.

Nodes related to matched nodes which might be helpful:

Sleep efficiency is related to Exercise:
Regular exercise is consistently linked to higher sleep efficiency, as it promotes deeper and more restorative sleep. This relationship is well-documented, with exercise serving as a non-pharmacological intervention for sleep disorders.

Sleep efficiency :
description: Sleep Efficiency (SE) is the percentage ... [truncated] |

| Response | Response | Response |
|---|---|---|
| Based on your exercise data from the past 7 days (November 15-21, 2019), the most frequent types of exercise you engaged in were:

1. **Walking** (11 sessions) - This was your dominant activity, with multiple daily walks logged both as auto-detected and tracker-recorded sessions. These walks typically lasted 20-40 minutes with moderate intensity (average heart rate 92-159 bpm).

2. **Treadmill** (1 session on November 19) - A single 26-minute vigorous session with higher intensity (average HR 154 bpm).

3. **General Workout** (1 session on November 19) - A 44-minute mixed-intensity session logged as "Workout" with strength training characteristics (lower step count but elevated HR zones).

Key observations:
- Walking accounted for 85% of all logged exercises
- No exercise was recorded on November 16
- The treadmill session was your most intense cardiovascular activity
- The "Workout" session suggests possible strength training

Note: Data shows consistent walking habits but limited variety in exercise types during this period. The treadmill and workout sessions indicate some cross-training efforts. | Based on your exercise data from the past 7 days (November 15-21, 2019), the most frequent types of exercise you engaged in were:

1. **Walking** (11 sessions) - This was your dominant activity, appearing nearly every day with multiple sessions. The walks typically lasted 20-40 minutes with moderate intensity (average heart rate 97-115 bpm).

2. **Treadmill** (1 session on Nov 19) - A single 26-minute vigorous session with higher heart rate (154 bpm average).

3. **General Workout** (1 session on Nov 19) - A 44-minute mixed-intensity session combining cardio and strength elements.

Key observations:
- Walking accounted for 85% of your recorded exercise sessions
- No strength training or flexibility exercises were logged in this period
- November 16 showed no recorded exercise activity

The data suggests a focus on cardiovascular exercise through walking, with one day incorporating more intense treadmill and workout sessions. For balanced fitness, you might consider adding strength training (2+ days/week) as recommended by health guidelines. | Based on your exercise data from the past 7 days (November 15-21, 2019), the most frequently engaged activities were:

1. **Walking** (9 instances) - This was your dominant activity, appearing nearly every day with multiple sessions. The walks typically lasted 20-30 minutes with moderate intensity (average heart rate 97-115 bpm).

2. **Treadmill** (1 instance on Nov 19) - A single 26-minute vigorous session with higher intensity (average HR 154 bpm).

3. **General Workout** (1 instance on Nov 19) - A 44-minute mixed-intensity session logged as "Workout" (average HR 133 bpm).

Key observations:
- Walking accounted for 82% of your logged exercise sessions
- No exercise was recorded on November 16
- Your most active day was November 19 with 4 separate exercise sessions
- Exercise intensity varied from light (walks) to vigorous (treadmill)

Your sleep efficiency remained consistently high (89-97%) throughout this period, which aligns with research showing positive sleep impacts from regular physical activity. |

**Example 2:** A multi-metric query, where examining multiple related metrics together provides a more comprehensive understanding.
**Query:** What is the overall relationship between HRV (RMSSD), steps taken, and lightly active time?
*Openness - 0.5*
*Query type - Metric Relationships*

| Base(Rank-3) – Context | Rag(Rank-2) – Context | WAG(Rank-1) – Context |
|---|---|---|
| Matched nodes:

HRV RMSSD :
sensor specific information if available:
1.description:
2.range:
3.unit:
Data:
— date — HRV RMSSD —
—:————————:—
— 2021-11-24 — 51.679 —
— 2021-11-25 — 47.082 —
— 2021-11-26 — 49.089 —
... [truncated]

Steps taken:
... [truncated]

Lightly active time:
sensor specific information if available:
... [truncated] | Matched nodes:

HRV RMSSD :
description: Heart Rate Variability Root Mean Square of Successive Differences (HRV RMSSD) is a statistical measure of heart rate variability (HRV)
... [truncated]
range: Normal HRV RMSSD values for adults at rest typically range from below 20 to over 70 milliseconds. However, individual values can vary significantly based on factors such as age, gender, fitness level, and overall health. Units: milliseconds (ms).
... [truncated]

Steps taken:
description: Steps Taken refers
... [truncated]

Lightly active time:
description: Lightly Active Time (LAT)
... [truncated] | Matched nodes:

HRV RMSSD :
description: Heart Rate Variability Root Mean Square of Successive Differences (HRV RMSSD) is a statistical measure of heart rate variability (HRV)
... [truncated]
range: Normal HRV RMSSD values for adults at rest typically range from below 20 to over 70 milliseconds. However, individual values can vary significantly based on factors such as age, gender, fitness level, and overall health. Units: milliseconds (ms).
... [truncated]

Nodes related to matched nodes which might be helpful:
Steps taken :
description: Steps Taken refers
... [truncated]

Lightly active time :
description: Lightly Active Time (LAT)
... [truncated] |
| **Response** | **Response** | **Response** |
| Based on the provided data from November 2021 to January 2022, the relationship between HRV RMSSD, steps taken, and lightly active time shows the following patterns:

1. **General Trends**:
- Extreme step counts (>20,000) often cor- - Higher HRV RMSSD (indicating better recovery/stress resilience) tends to coincide with days of moderate step counts (10,000-17,000) and lightly active time (150-200 minutes). Examples: 2021-12-08 (HRV: 62.585, Steps: 11,727, Light activity: 156 min).
- Extreme step counts (>20,000) often correlate with lower HRV (e.g., 2021-12-29: 22,634 steps → HRV 47.801; 2021-12-31: 27,943 steps → HRV 48.292).

2. **Notable Exceptions**:
- Some high-activity days maintained good HRV (e.g., 2021-12-17: 20,005 steps, 262 min light activity, HRV 50.73).
- The lowest HRV values (e.g., 2021-12-26: 34.656) occurred after consecutive high-activity days, suggesting possible cumulative fatigue.
... [truncated] | 1. **General Relationship**:
- HRV RMSSD (a marker of parasympathetic/vagal activity) typically shows an inverted-U relationship with physical activity:
... [truncated]
2. **Data Observations** (Nov 2021-Jan 2022):
- On high step days (>15,000 steps) with substantial lightly active time (>200 min):
- HRV often remained stable (e.g., Dec 17: 20,005 steps, 262 min LAT → HRV 50.73 ms)
- Some high-activity days showed HRV dips (Dec 25: 16,722 steps → HRV 38.93 ms)
- Lower activity days (<5,000 steps) showed variable HRV:
- Some low (Dec 12: 1,688 steps → HRV 52.82 ms)
- Some high (Dec 7: 10,245 steps → HRV 56.27 ms)

3. **Key Patterns**:
- No strict linear correlation exists in this dataset
- HRV appears more sensitive to recovery factors (sleep, stress) than to activity volume alone
- The individual maintains mid-range HRV (45-55ms) across most activity levels, suggesting good autonomic resilience
... [truncated] | 1. **General Trends**:
- Higher HRV RMSSD (indicating better recovery/stress resilience) tends to coincide with days of moderate step counts (10,000-15,000 steps) and balanced lightly active time (150-200 minutes).
- Extreme activity days (e.g., Dec 31: 27,943 steps) often correlate with lower HRV the following day (Jan 1: 37.97 ms), suggesting potential overexertion.
2. **Optimal Range Observations**:
- The highest HRV readings (Dec 8-10: 62-64 ms) occurred with:
... [truncated]
- This suggests a "sweet spot" where substantial but not excessive movement supports autonomic balance.
3. **Recovery Dynamics**:
- Days with <8,000 steps and <120 light activity minutes (e.g., Dec 3, Dec 12) often preceded HRV rebounds, indicating these may function as recovery days.
4. **Notable Exceptions**:
- December 25-26 shows an inverse pattern where high steps (16,722-10,128) coincided with very low HRV (38.93-34.66 ms), possibly indicating holiday stress or sleep disruption overriding activity benefits.
**Key Insight**: Moderate daily movement (10,000-15,000 steps with 150-200 light activity minutes) appears most supportive of healthy HRV levels in
... [truncated] |

**Example 3:** A question where the primary node lacks direct data. By incorporating the KG, we can identify related nodes and leverage other metrics to approximate the condition of the queried metric.

**Query:** How is my engagement levels over the past week?

*Openness - 0.8*

*Query type - Exploratory Analysis*

| Base(Rank-2) – Context | Rag(Rank-3) – Context | WAG(Rank-1) – Context |
|---|---|---|
| Matched nodes:

data: No data | Matched nodes:

data: No data | Matched nodes:
data: No data

Nodes related to matched nodes which might be helpful:
... [truncated]

Steps taken :
description: Steps Taken refers
... [truncated]
Data:
— date — Steps taken —
—:——:—
— 2020-01-13 — 2873 —
— 2020-01-14 — 284 —
... [truncated]
Energy expenditure:
description: Energy Expenditure (EE) refers to ... [truncated]

Mental fatigue :
description: Mental fatigue is a
... [truncated]

Active time :
description: Active Minutes refer to... [truncated] |

| Response | Response | Response |
|---|---|---|
| Answer: I cannot determine factors influencing your engagement levels because no relevant data was available for analysis over the past week. To provide meaningful insights, I would need access to metrics such as:
- Sleep quality/duration
- Physical activity levels
- Heart rate variability (HRV)
- Cognitive load/stress indicators

Without this foundational data, I can only suggest general considerations like sleep hygiene, stress management, and hydration that commonly affect engagement. Would you like to discuss methods for tracking these metrics?

(Data limitation acknowledged: No retrievable engagement-related biomarkers) | Answer: I cannot determine factors influencing your engagement levels due to unavailable data. To provide meaningful analysis, I would need:

1. **Engagement Metrics**: Tracked behaviors (e.g., workout consistency, task completion rates)
2. **Correlative Data**: Sleep quality, stress markers (HRV), or activity logs for the past 7 days
Without this baseline, I recommend:
- Enabling granular tracking for engagement-related metrics
- Checking device/sensor connectivity to ensure data collection

Specify any available proxies (e.g., screen time, step count) for alternative assessment.

*(Limitation: Empty response from knowledge retrieval)* | Based on the available data from the past week, several key factors appear to be influencing your engagement levels:
1. **Physical Activity Patterns**:
- Your step count showed significant variation (284 to 30,406 steps/day)
... [truncated]
2. **Energy Expenditure**:
- Caloric burn ranged from 1,997-5,012 kcal/day (+1.30 deviation), with
... [truncated]
3. **Mental Fatigue**:
- Your self-reported fatigue scores showed inverse patterns to activity levels. Notably, your lowest fatigue score (1) coincided with your highest activity day, while higher fatigue (4) occurred during less active periods.
**Key Observations**:
- The data suggests a strong temporal association between physical activity metrics (steps, active minutes, energy expenditure) and reduced mental fatigue, which collectively influence engagement capacity.

- Your most engaged days (e.g., 2020-01-19) show the "perfect storm" of high activity (+30k steps, 471 active mins, 5,012 kcal) coupled with minimal mental fatigue (score=1).

- The 1.22–1.46 deviations in physical metrics indicate this was an unusually active week compared to your baseline, which may explain heightened engagement.

... [truncated] |

**Example 4:** A case where incorporating additional metrics provides deeper insights beyond the primary data, enabling a more comprehensive understanding of the queried condition.

**Query:** How does my bedtime start time variability over the past 30 days compare to recommended sleep hygiene guidelines?

*Openness - 0.6 Query type - Comparative Insight*

| Base(Rank-2) – Context | Rag(Rank-3) – Context | WAG(Rank-1) – Context |
|---|---|---|
| Matched nodes:

Bedtime start time :
sensor specific information if available:
1.description:
2.range:
3.unit:
Data:
— date — Bedtime start time —
—:————:——————
— 2020-02-10 — 2020-02-09 21:54:30 —
— 2020-02-11 — 2020-02-10 23:05:00 —
... [truncated] | Matched nodes:

Bedtime start time :
description: Bedtime Start
... [truncated]
range: Recommended bedtime start times vary by age group: School-age children (6-13 years): 8:00 - 9:30 p.m., Teens (14-17 years): 9:00 - 10:30 p.m., Young adults (18-25 years): 8:00 - 12:00 p.m., Adults (26-64 years): 8:00 - 12:00 p.m. Consistency is more important than the exact time.
recommendation: To improve sleep hygiene, maintain a consistent bedtime and wake-up schedule, create a relaxing pre-sleep routine, optimize the sleep environment (e.g., comfortable bedding, reduced light and noise), avoid stimulating activities or heavy meals before bed, and seek professional help if sleep issues persist.
sensor specific information:
... [truncated] | Matched nodes:

Bedtime start time :
description: Bedtime Start
... [truncated]

Nodes related to matched nodes which might be helpful:

Bedtime end time :
description: Bedtime End Time refers
... [truncated]
Mood :
description: Mood refers to
... [truncated]

Asleep duration :
description: Total Sleep Duration (TSD)... [truncated] |

| Response | Response | Response |
|---|---|---|
| Answer: Your bedtime start time variability over the past 30 days shows significant fluctuations, ranging from 21:26 to 01:26, with a standard deviation of approximately 1.5 hours. This exceeds the recommended sleep hygiene guideline of maintaining bedtime consistency within a 30-minute window (National Sleep Foundation). Key observations:

1. **Pattern Analysis**:
- 40% of bedtimes occurred after midnight
- Only 20% fell within a consistent 30-minute window of your median bedtime (22:53)

2. **Guideline Comparison**:
- Your variability is 3x greater than the recommended threshold
- Late bedtimes (post-23:30) correlate with documented circadian rhythm disruption risks

3. **Data Limitations**:
- 1 missing data point (2020-03-03)
- No baseline comparison available

Recommendation: Gradually adjust bedtime toward a consistent target (e.g., 22:30-23:00) to align with chronobiology research showing improved sleep quality with regular schedules. Consider implementing a 30-minute pre-bed routine to reinforce circadian cues. | Your bedtime start time variability over the past 30 days shows significant inconsistency, which deviates from recommended sleep hygiene guidelines. Here's the analysis:

1. **Variability Pattern**:
- Bedtimes range widely from 21:26 to 01:26 (a 4-hour window)
- Frequent late bedtimes (past midnight on 8 occasions)
- No consistent pattern emerges from the data

2. **Guideline Comparison**:
- Sleep hygiene emphasizes consistency (±30 minutes is ideal)
- Your variability exceeds this by 8x the recommended threshold
- Late bedtimes may indicate circadian rhythm disruption

3. **Data Observations**:
- 27% of nights had bedtimes after midnight
- Only 3 nights fell within a 1-hour window (22:30-23:30)
- One missing data point (2020-03-03)

... [truncated] | Your bedtime start time variability over the past 30 days shows significant inconsistency, which deviates from sleep hygiene recommendations emphasizing consistency. Key observations:

1. **Variability Pattern**:
- Bedtime ranges from 21:26 to 01:26 (4-hour window), with frequent 1-hour day-to-day shifts
- 40% of nights had bedtimes after midnight (vs. recommended 8pm-12am window for adults)

2. **Impact on Sleep Metrics**:
- Later bedtimes correlate with shorter sleep duration (e.g., 01:26 bedtime → 5.4h sleep vs 21:26 → 9h sleep).
- Mood data (where available) shows better scores (4/5) on days following earlier bedtimes.

3. **Guideline Comparison**:
- Sleep hygiene recommends ¡30min variability in bedtime
- Your average variability exceeds 90min, with 7 instances of 2-hour differences between consecutive nights
... [truncated] |

**Example 5:** A case where incorporating additional metrics provides deeper insights beyond the primary data, enabling a more comprehensive understanding of the queried condition.

**Query:** Have there been any noticeable patterns in my mental fatigue over the past week?

*Openness - 0.5*

*Query type - Trend Analysis*

| Base(Rank-3) – Context | Rag(Rank-2) – Context | WAG(Rank-1) – Context |
|---|---|---|
| Matched nodes:

Mental fatigue :

sensor specific information if available:
1.description: self-report fatigue score
2.range: 1-5 scale. Score 3 is normal, and 1-2 are scores below normal, and 4-5 are scores above normal
3.unit:
Data:
— date — Mental fatigue —
—:——————————:—
— 2020-03-02 — 3 —
— 2020-03-03 — 2 —
— 2020-03-04 — 3 —
— 2020-03-05 — 4 —
— 2020-03-06 — 2 —
— 2020-03-07 — 4 —
— 2020-03-08 — 2 —
No deviation from baseline recorded for the recent 7-day period. | Matched nodes:

Mental fatigue :

description: Mental fatigue is a psychobiological state characterized by feelings of tiredness and reduced cognitive efficiency, typically resulting from prolonged periods of demanding mental activity. It manifests as difficulty concentrating, slower mental processing, and increased errors in task performance. Mental fatigue can also present physical symptoms such as headaches, stomach aches, and sleep disturbances. It is commonly assessed through self-reported scales and behavioral performance metrics.
range: Mental fatigue is typically measured using subjective scales, such as the Mental Fatigue Scale (MFS), ... [truncated] | Matched nodes:

Mental fatigue :
description: Mental fatigue... [truncated]

Nodes related to matched nodes which might be helpful:

Total sleep duration :
description: Total Sleep Duration (TST) refers to ... [truncated]

Asleep duration :
description: Total Sleep Duration (TSD) refers to ... [truncated] |

| Response | Response | Response |
|---|---|---|
| Answer: Over the past week, your mental fatigue scores show variability with no clear upward or downward trend. The pattern alternates between normal (score 3) and deviations in both directions:
- **Below normal** (scores 2) on 3/03, 3/06, and 3/08
- **Above normal** (scores 4) on 3/05 and 3/07

No baseline deviation was recorded, suggesting these fluctuations may represent typical variation rather than a sustained shift. For actionable insights, consider correlating these scores with sleep, stress, or activity data if available.

*Limitation: Self-reported scores may not capture all contributing factors.* | Your mental fatigue data over the past week (March 2-8, 2020) shows a fluctuating pattern with scores alternating between normal (score 3) and elevated levels (scores 4-5). Key observations:

1. **Pattern**:
- 3 days at normal baseline (score 3)
- 2 days with elevated fatigue (score 4)
- 2 days with below-normal reports (score 2)

2. **Notable Peaks**:
- March 5th and 7th showed elevated fatigue (score 4), indicating days of significantly increased mental tiredness

3. **Comparison**:
- This variation falls within your normal range (no statistical abnormality detected)

The alternating pattern suggests possible workload variability or recovery periods between demanding cognitive activities. The two elevated fatigue days may warrant examination of specific activities or stressors occurring on March 5th and 7th.

Recommendation: Consider implementing the fatigue management strategies mentioned (timeboxing, scheduled breaks) particularly on days following elevated fatigue scores to support recovery. | ### Analysis of Mental Fatigue Patterns (Past Week):

**1. Mental Fatigue Scores:**
- Your self-reported fatigue scores (1-5 scale) show variability:
- **Higher fatigue days (score 4):** March 5th and 7th
... [truncated]
**2. Sleep Correlations:**
- **Total Sleep Duration:**
- Lowest sleep (351 min on March 3rd) preceded a low fatigue day (March 3rd score 2)
- Highest sleep (594 min on March 7th) coincided with high fatigue (March 7th score 4)
- This inverse pattern suggests sleep duration alone doesn't explain your fatigue levels.
- **Asleep Duration:**
- Similar pattern observed, with no clear linear relationship to fatigue scores.

**3. Key Observations:**
- Your highest fatigue days (March 5th and 7th) followed days with *above-average* sleep duration (560 min and 568 min respectively on prior nights).

- Lowest fatigue occurred after your *shortest* sleep night (March 3rd after 351 min sleep).

- This may indicate:
- Potential sleep quality issues (e.g., disrupted sleep architecture despite longer duration)
- Non-sleep factors influencing fatigue (e.g., stress, cognitive load)
- Delayed fatigue effects (e.g., fatigue manifesting 1-2 days after sleep changes)

... [truncated] |

**Example 6:** A case where the query involves non-numerical metric.
**Query:** What key insights can be derived from my lifelog data over the past 7 days?
*Openness - 0.9*
*Query type - Exploratory Analysis*

| Base(Rank-2) – Context | Rag(Rank-3) – Context | WAG(Rank-1) – Context |
|---|---|---|
| Matched nodes:

Lifelog :

sensor specific information if available:
1.description:
2.range:
3.unit:
Data:
— date — Lifelog —
—:—————:——————————
————————————————
— 2019-11-13 — I suffered from injuries: {'left_elbow': 'minor', 'right_shoulder': 'minor'} —
No deviation from baseline recorded for the recent 7-day period. | Matched nodes:

Lifelog :

description: A lifelog is a comprehensive digital record of an individual's daily life, capturing events, activities, and interactions in varying levels of detail. It serves as a personal archive that can be used for memory retrieval, behavioral analysis, and health monitoring. Lifelogging systems often utilize wearable devices, smartphones, and other sensors to collect data, which can be analyzed to identify patterns, predict behaviors, and support decision-making in healthcare and personal development.
... [truncated]
— 2019-11-13 — I suffered from injuries: {'left_elbow': 'minor', 'right_shoulder': 'minor'} —
Abnormality of recent 7 days compared to individual's average: No abnormality level | Matched nodes:

Lifelog :

description: A lifelog is a comprehensive digital record of an individual's daily life,
... [truncated]

Nodes related to matched nodes which might be helpful:

Deep sleep duration :

... [truncated]
— date — Deep sleep duration —
—:—————————:—
— 2019-11-07 — 40 —
— 2019-11-08 — 73 —
— 2019-11-09 — 35 —
— 2019-11-10 — 76 —
— 2019-11-11 — 67 —
— 2019-11-12 — 4 —
— 2019-11-13 — 28 —
Moderately active time :

... [truncated]

Asleep duration :

... [truncated]

Total sleep duration :

... [truncated] |
| **Response** | **Response** | **Response** |
| Answer: Your lifelog data for the past 7 days shows no recorded health metrics (e.g., HRV, heart rate, or activity levels) as all entries except one are marked 'nan' (not available). The only notable event was on 2019-11-13, when you reported minor injuries to your left elbow and right shoulder.

**Key Insights:**
1. **Data Gap**: No measurable health data was captured for 6 out of 7 days, limiting trend analysis.
2. **Injury Context**: The minor injuries on 2019-11-13 may warrant monitoring for recovery (e.g., pain duration, mobility changes), but no follow-up data is available.

**Recommendations:**
- Verify sensor functionality to ensure future data collection.
- Log additional details about injuries (e.g., cause, symptoms) to assess impact.
- If injuries persist beyond typical healing time (7–10 days for minor soft-tissue injuries), consult a healthcare provider.

*Limitation*: Insights are constrained by missing data. No baseline comparisons or longitudinal trends can be derived. | Answer: Your lifelog data for the past 7 days shows limited recorded activity, with only one notable entry on **2019-11-13** indicating minor injuries to your left elbow and right shoulder. No other health metrics, activities, or physiological data were captured during this period.

### Key Insights:
1. **Data Gaps**: The lifelog contains mostly empty entries ('nan'), suggesting inconsistent data collection or potential device/sensor issues.
2. **Injury Record**: The minor injuries logged on 2019-11-13 may warrant monitoring for recovery progress, but no follow-up data is available.
3. **No Abnormalities**: The system detected no deviations from your baseline, but this assessment is limited due to sparse data.

### Recommendations:
- Verify device/sensor functionality to ensure continuous data capture.
- Expand tracked metrics (e.g., activity levels, sleep, HRV) for more actionable insights.
- Log updates on injury recovery if symptoms persist.

*Note: Conclusions are constrained by insufficient data granularity.* | Here are the key insights derived from your lifelog data over the past 7 days:

1. **Injury Event**: On 2019-11-13, you reported minor injuries to your left elbow and right shoulder. This may correlate with the observed fluctuations in sleep and activity metrics.
2. **Sleep Patterns**:
- **Deep Sleep**: Varied significantly (4-76 minutes), with a notable drop to just 4 minutes on 2019-11-12. The 7-day average deviates +1.18 SD from your baseline, suggesting unusual instability.
- **Total Sleep Duration**: Ranged from 220-478 minutes (~3.7-8 hours), with the shortest duration coinciding with the day of lowest deep sleep (2019-11-12).
3. **Activity Levels**:
- Moderately active time showed a progressive increase (14-49 minutes/day), peaking on the injury reporting day (2019-11-13). The 7-day average is 0.83 SD above your baseline.
4. **Critical Observations**:
- The injury date (2019-11-13) coincided with both your highest activity level (49 minutes) and suboptimal deep sleep (28 minutes).

- The most extreme sleep disruption (2019-11-12) preceded the injury report, suggesting possible fatigue as a contributing factor.

... [truncated] |

## K  PROMPTS

### Prompt 1: `QueryGen_single`

```
"""
Generate diverse, clinically relevant questions about health metrics from wearable data.

INPUT FORMAT (Array of metric objects):
[
    {
        "id": "<unique_id>",
        "name": "<metric_name>", # The health metric being analyzed
        "description": "<definition>", # Clinical definition of the metric
        "date": "<YYYY-MM-DD>",
        "time_granularity": "<1|7|14|30|60|all>", # Time period covered
        "abnormality_level": "<low|medium|high>", # Deviation from user's baseline
    },
    ...
]

OUTPUT FORMAT (Array of questions - one per input metric):
[
    {
        "id": "<matching_input_id>",
        "question": "<clear, time-bound phrasing>",
        "question_type": "<one of: General Knowledge | Data Retrieval | Trend Analysis |
            Comparative Insight | Anomaly Detection | Actionable Advice | Exploratory
            Analysis>",
        "openness": <0.0-1.0>, # 0.0=closed, 1.0=open-ended
    },
    ...
]
QUESTION FRAMEWORK:
1. **General Knowledge** (Openness: 0.2-0.4)
   - Definitions, benchmarks, normal ranges
   - Example: "What's considered a healthy range for [metric]?" "What is [metric]?"

2. **Data Retrieval** (Openness: 0.1-0.3)
   - Specific time-bound numerical queries
   - Example: "What was my [metric] yesterday?" "What was my max/min/average [metric] this
     week?"

3. **Trend Analysis** (Openness: 0.4-0.6)
   - Patterns over days/weeks/months
   - Example: "Identify any trends in my [metric] over the last 30 days." "Summarize my
     [metric] for the past month."

4. **Comparative Insight** (Openness: 0.5-0.7)
   - Time-period comparisons
   - Example: "How does this week's [metric] compare to last week?"

5. **Anomaly Detection** (Openness: 0.6-0.8)
   - Statistical outliers
   - Example: "Were there unusual [metric] spikes in this month?"

6. **Actionable Advice** (Openness: 0.3-0.5)
   - Data-driven recommendations
   - Example: "What adjustments could improve my [metric]?"

7. **Exploratory Analysis** (Openness: 0.7-1.0)
   - Multi-factor investigations
   - Example: "Do you think I am stressed recently?" "I'm feeling really tired today. do you
     know why?" "Why might I be feeling tired despite sleeping 8 hours?"

GENERATION RULES:
1. Time binding:
   - Map granularity to natural terms:
     1   "today"
     7   "past 7 days"
     14  "past 14 days"
     30  "past 30 days"
     60  "past 60 days"
     all "overall"
2. Blend concrete and exploratory questions per category:
   - 40% objective (openness 0.4)
   - 30% moderate (0.4 < openness <0.7)
   - 30% open-ended (0.7)
3. Prevent overlap between categories
4. For medium/high abnormalities, prioritize generating high openness questions
5. Exactly 1 output question per input group

EXAMPLES:

INPUT:
[
    {
        "id": "m001",
        "name": "Inactive time",
        "description": "The amount of time a user is inactive, measured in minutes",
        "date": "2020-01-01",
        "time_granularity": "1",
```

```
          "abnormality_level": "low",
        },
        {
          "id": "m002",
          "name": "total sleep time",
          "description": "The total amount of time a user spends in sleep",
          "date": "2020-02-02",
          "time_granularity": "14",
          "abnormality_level": "high",
        },
        ...
    ]
]
OUTPUT:
[
    {
        "id": "m001",
        "question": "What was my inactive time today?",
        "question_type": "Data Retrieval",
        "openness": 0.1,
    },
    {
        "id": "m002",
        "question": "How does my total sleep time over the past 14 days compare to the previous
            period?",
        "question_type": "Comparative Insight",
        "openness": 0.7,
    },
    ...
]

"""
```

Prompt 2: QueryGen_multiple

```
"""
Generate clinically relevant questions from wearable data, with each question containing 2-3
    metrics.

INPUT FORMAT (Array of metric objects):
[
    {
        "id": "<unique_id>",
        "metrics":[
            {
                "name": "<metric_name_1>", # The health metric being analyzed
                "description": "<definition>", # Clinical definition of the metric
            },
            {
                "name": "<metric_name_2>", # The health metric being analyzed
                "description": "<definition>", # Clinical definition of the metric
            },
            ...
        ],
        "date": "<YYYY-MM-DD>",
        "time_granularity": "<1|7|14|30|60|all>", # Time period covered
    },
    ...
]
OUTPUT FORMAT (Array of questions - one per input):
[
    {
        "id": "<matching_input_id>",
        "question": "<clear, time-bound phrasing>",
        "question_type": "<one of: Metric Relationships | Contextual Queries>",
        "openness": <0.0-1.0>, # 0.0=closed, 1.0=open-ended
    },
    ...
]
QUESTION FRAMEWORK:
1. **Metric Relationships** (Openness: 0.4-0.6)
  - Example: "Does [metric1] relate to [metric2] trends for the past 30 days?"

2. **Contextual Queries** (Openness: 0.5-0.7)
  - Example: "Do [metric1] spikes follow days with high [metric2]?" "Is there a pattern in my
      [metric1] on days I have a higher [metric2] for the past week?"

GENERATION RULES:
1. Time binding:
  - Map granularity to natural terms:
    1    "today"
    7    "past 7 days"
    14   "past 14 days"
    30   "past 30 days"
    all  "overall"
2. Each question must reference metrics from the input list
3. Exactly 1 output question per input group
EXAMPLES:
INPUT:
[
    {
        "id": "001",
```

```
2106
2107          "metrics": [
2108              {
                      "name": "resting_heart_rate",
2109                  "description": "Beats per minute at complete rest"
2110              },
                  {
2111                  "name": "sleep_duration",
                      "description": "Total minutes of sleep per night"
2112              }
2113          ],
              "date": "2023-11-15",
2114          "time_granularity": "30"
          }
2115     ]
        OUTPUT:
2116     [
            {
2117            "id": "001",
               "question": "How does my resting heart rate vary with sleep duration for the past 30
2118                   days?",
               "question_type": "Metric Relationships",
2119            "openness": 0.5,
            }
2120     ]

2121     """
2122
```

### Prompt 3: Context

```
2123     """
2124     You are a clinical expert in wearable sensor measurements.
         The goal is to generate a knowledge graph that connects multimodal wearable data (e.g., sleep
2125             metrics, activity levels, and self-reported affect)
         This graph will serve as a key resource for a retrieval-augmented generation process in an
2126             LLM, supporting insight discovery, outcome prediction, and personalized intervention
                 design.
2127     Process:
         step 1 Initial Node Creation:
2128     given a comprehensive list of health metrics commonly measurable by wearable devices,
             generate a node representation for each metric.
2129
2130     step 2 Relationship Mapping:
         given all the nodes, determine the relationships between each pair of nodes and create edges
2131             between them.
2132
         step 3 New Metric Integration:
2133     given a list of new wearable health metrics and all existing nodes.
         for each metric, you will need to check against existing graph nodes to identify potential
2134             duplicates,and merge if a match was found.
2135
         step 4 Graph Extension:
2136     the remaining new metrics from step 3 will be added to the graph as new nodes and edges will
             be created to connect them to the existing nodes.
2137     """
2138
2139
2140
2141
2142
2143
2144
2145
2146
2147
2148
2149
2150
2151
2152
2153
2154
2155
2156
2157
2158
2159
```

Prompt 4: NodeGen

```
"""
-Goal-
Generate a standardized node representation for a wearable sensor measurement that
    synthesizes available information with your own clinical knowledge.
be aware that some reference data may not be related to the entity, so you should be careful
    to filter out the irrelevant information but the provided data is important and should
    always be used.
-Input Format-
{
    "entity_name": "<measurement_name>",

    // source: Device/sensor documentation
    "provided_description": "<provided_description>",
    "provided_range": "<provided_range>",

    // source: web search
    "web_description": "<web_search_results>",
    "value_range": "<ranges>",
    "recommendations": "<guidelines>"

    // source: UMLS
    "umls_description": "<umls_definition>",

}

-Output Format-
{
    "name": "<standardized_name>",
    "description": "<comprehensive_description>",
    "range": "<range_info_or_None>",
    "recommendations": "<guidelines_or_None>"
}

-Guidelines-
1. Name:
   - Use standardized medical terminology
   - Keep concise but clear
   - Include common abbreviation if applicable

2. Description (Required):
   - What is being measured
   - How it's measured
   - Clinical significance
   - Relationship to health outcomes

3. Range (if applicable):
   - Normal ranges for different demographics
   - Units of measurement
   - Alert thresholds
   - Output "None" if not applicable(e.g. gesture recognized does not have a range)

4. Recommendations (if applicable):
   - Evidence-based
   - Actionable
   - Context-aware
   - Output "None" if not applicable (e.g. there is not such a recommendation for improvement
       for the entity, like gesture)

-Examples-
Example 1:
Input:
{
  "entity_name": "Energy expenditure",
  "provided_description": "Energy consumption caused by the physical activity of the day.",
  "provided_range": "range: None unit: kcal",
  "web_description": "As people pursue activities at multiple locations, trips are produced
      between successive activity locations. Patterns formed by trips over a period, such as
      a day, are called 'travel patterns.\nReference 0: Pattern means two or more acts
      occurring over a period of time, however short, ...
  "umls_description": "",
  "value_range": "\nReference 0: A travel pattern refers to the classification of daily travel
      behaviors based on factors such as the number of trips and total travel
      time.\nReference 1: Long-distance trips are journeys of more than 50 miles from home to
      the furthest destination.\nReference 2: New MIT research confirms peoplec
  ......[truncated].......

"""
```

Prompt 5: EdgeGen

```
"""
-Goal-
Analyze relationships between pairs of wearable health metrics and generate standardized edge
    representations.
Use provided descriptions, web search results, and your own clinical knowledge to determine
    meaningful relationships.
be aware that some reference information may not be accurate or related to the entity, so you
    should be careful to filter out the irrelevant information.
-Input Format-
[
    {
    "id": "<id>",
    "entity_1_name": "<entity_1_name>",
    "entity_1_description": "<entity_1_description>",
    "entity_2_name": "<entity_2_name>",
    "entity_2_description": "<entity_2_description>",
    "web_search_results": "<relevant_search_results>",
    }
    ...
]

-Output Format-
[
    {
        "id": str,
        "entity_1_name": str,
        "entity_2_name": str,
        "relationship": {
            "description": str, # Detailed explanation of the relationship
            "strength": float, # 0.1 to 1.0
            "confidence": str # "high", "medium", or "low" based on evidence quality
        },
    },
    ...
]

RELATIONSHIP SCORING:
- Strong (0.7-1.0):
  * Clear scientific evidence
  * Direct causal or strong correlational relationship
  * Well-documented in medical literature

- Moderate (0.3-0.6):
  * Some scientific evidence
  * Indirect or secondary relationship
  * Limited but consistent documentation

- Weak (0.1-0.2):
  * Limited or circumstantial evidence
  * Indirect relationship with multiple variables
  * Inconsistent documentation

- Not Related (<0.1):
  * Exclude from output
  * No meaningful connection
  * No supporting evidence

-Examples-
Example 1:
Input:
[
    {
    "id": "1",
    "entity_1_name": "Heart rate",
    "entity_1_description": "The number of heartbeats per unit of time, usually expressed as
        beats per minute."
    "entity_2_name": "Blood pressure",
    "entity_2_description": "The pressure of the circulating blood against the walls of the
        blood vessels.",
    "web_search_results": "Elevated heart rate is associated with elevated blood pressure,
        increased
    ......[truncated].......
"""
```

Prompt 6: Merge

```
"""
-Goal-
Analyze a new node against existing nodes to identify potential duplicates in a wearable
     health knowledge graph.

GUIDELINES FOR COMPARISON:
1. Semantic Analysis:
   - Look beyond exact text matches
   - Consider medical synonyms and related terms
   - Evaluate contextual meaning in healthcare

2. Description Analysis:
   - Identify overlapping concepts
   - Consider complementary information
   - Evaluate scope and specificity

3. Scoring Criteria:
   0.0-0.3: Clearly different concepts
   0.4-0.6: Related but distinct
   0.7-0.8: Highly similar
   0.9-1.0: Virtually identical

4. Only return the node if you think it is a duplicate of an existing node.

INPUT FORMAT:
{
    "input_name": "new node name",
    "input_description": "new node description",
    "references": [
        {
            "name": "existing node 1 name",
            "description": "existing node 1 description"
        },
        {
            "name": "existing node 2 name",
            "description": "existing node 2 description"
        }
        ...
    ]
}

OUTPUT FORMAT:
[
    {
        "input_name": "new node name",
        "reference_name": "matched existing node name",
        "similarity_score": <float 0-1>,
        "same_concept": <boolean>,
        "reasoning": "clear explanation of similarity assessment and why the nodes are the same
            or different"
    }
    ...
]

EXAMPLE :
Input:
{
    "input_name": "steps",
    "input_description": "Total number of steps registered during the day.",
    "references": [
    {
      "name": "Steps taken",
      "description": "Steps Taken refers to the total number of steps registered by a wearable
            device over a given period, typically a day. It is a key metric for assessing
            physical activity levels, with higher step counts generally associated with better
            cardiovascular health, weight management, and overall fitness. Wearable devices
            track steps using accelerometers or gyroscopes to detect motion and count steps
            based on movement patterns."
    },
    ......[truncated].......
    ]
}
Output:
[
    {
      "input_name": "steps",
      "reference_name": "Steps taken",
      "similarity_score": 0.95,
      "same_concept": true,
      "reasoning": "Both nodes refer to the total number of steps registered by a wearable
            device over a given period, typically a day. The descriptions are nearly identical,
            with both emphasizing the use of accelerometers or gyroscopes to detect motion and
            count steps. The terms 'steps' and 'steps taken' are semantically equivalent in this
            context."
    }
]

"""
```

### Prompt 7: Query_base

```
"""
## CORE OBJECTIVE
Analyze health queries through structured function calls to external knowledge retrieval APIs,
    synthesizing results into evidence-based responses through systematic analysis of
    retrived context.

## EXECUTION FRAMEWORK

1. **QUERY DECOMPOSITION**
   - **Key Entities**: Identify health metrics (e.g., HRV, heart rate)
   - **Temporal Scope**:
    - Default: Past 7 days
    - Explicitly stated periods override default

2. **KNOWLEDGE RETRIEVAL**
   - Primary Entity Matching: Fetch data for core health metric.
   - Contextual Filtering: Apply time-based constraints.

3. **ANALYSIS**
   - Cross-reference data with medical best practices.
   - Highlight trends, anomalies, or gaps.

4. **RESPONSE GENERATION**
   - Requirements:
    - Ground all claims in evidence
    - Acknowledge data limitations
    - For unanswerable queries: Specify missing data

## OUTPUT FORMAT
Answer: Concise response with integrated insights.
"""
```

### Prompt 8: Query_wag

```
"""
## CORE OBJECTIVE
Analyze health queries through structured function calls to external graph traversal APIs,
    synthesizing results into evidence-based responses through systematic analysis of
    entities, relationships, and multimodal connections.

## EXECUTION FRAMEWORK

1. **QUERY DECOMPOSITION**
   - **Key Entities**: Identify primary subjects/measurements (e.g., HRV, heart rate)
   - **Temporal Scope**:
    - Default: Past 7 days
    - Explicitly stated periods override default
   - **Openness Score** (0.0-1.0):
    | Score Range | Search Strategy | Examples |
    |-------------|-------------------------------|-------------------------------|
    | 0.0-0.3 | Narrow focus on exact matches | "Optimal HRV range?" (0.0) |
    | 0.4-0.7 | Balanced entity+relationships | "HR comparison week/week?" (0.5) |
    | 0.8-1.0 | Broad multimodal exploration | "Why elevated heart rate?" (0.9) |

2. **GRAPH TRAVERSAL**
   - Primary entity matching
   - Relationship expansion proportional to openness score
   - Contextual data retrieval with temporal filtering

3. **MULTIMODAL ANALYSIS**
   - Cross-reference data types:
    * Physiological (HRV, HR)
    * Environmental (sleep, activity)
    * Subjective (user notes)
   - Identify:
    - Consistent corroborating evidence
    - Conflicting indicators
    - Temporal patterns

4. **RESPONSE GENERATION**
   - Requirements:
    - Ground all claims in evidence
    - Acknowledge data limitations
    - For unanswerable queries: Specify missing data
   - Prioritize:
    - Direct correlations > inferred relationships
    - User-specific context > general knowledge

## OUTPUT FORMAT
Answer: Concise response with integrated insights.
"""
```

## Prompt 9: Eval

```
"""

You are an expert in clinical evaluation and human-centered AI systems. Your task is to
    evaluate and compare the quality of response generated by multiple methods to answer a
    user's health-related query based on their wearable data.

Each retrieval method provides a different set of contextual knowledge (e.g., entities,
    relationships, multimodal time-series patterns) intended to support answering the user's
    question. Your goal is to assess the quality of the response generated by each method.

-Input Format-
{
  "query": "<user's health question>",
  "methods": {
    "method_1": {
      "response": "<generated answer>"
    },
    "method_2": {
      "response": "<generated answer>"
    },
    ...
  }
}

-Evaluation Criteria-
Rank the methods from 1 (best) to N (worst) for each of the following dimensions:

1. **Insightfulness(most important)**: Does the response offer meaningful, actionable
    insights beyond the obvious?
2. **Relevance**: Is the response relevant and does it include novel information?
3. **Groundedness**: Are factual claims well-supported by the provided content or trusted
    sources?
4. **Personalization**: Does the response meaningfully incorporate the user's context (e.g.,
    wearable data)?
5. **Clarity**: Is the response clearly written, logically structured, and easy to understand
    for a non-expert?
6. **Absence_of_harmful_content**: Is the response free from misleading, unsafe, or
    inappropriate information?

Important Notes:

Do not assign the same rank to multiple methods unless they are truly indistinguishable in
    that dimension.

Rank relative to each other within the batch, not by absolute standards.

Lower rank numbers are better (1 = best performance for that criterion).

-Output Format-
Return a dictionary with evaluation scores per method:

{
  "method_1": {
    "Overall_quality": <1-N>,
    "Insightfulness": <1-N>,
    "Relevance": <1-N>,
    "Groundedness": <1-N>,
    "Personalization": <1-N>,
    "Clarity": <1-N>,
    "Absence_of_harmful_content": <0 or 1>
  },
  ...
}
"""
```

