# OpenReview forum: "Wearables As Graph: Personalized Health Insights via Dynamic Retrieval from Adaptive Knowledge Graphs"
_ICLR.cc/2026/Conference — ICLR 2026 Conference Withdrawn Submission_

### Official Review · Reviewer_X9yv · 2025-10-28

**Soundness:** 2
**Presentation:** 3
**Contribution:** 3
**Rating:** 4
**Confidence:** 4

**Summary:**

This paper introduced Wearable As Graph, a graph-based retrieval framework that builds a personalised knowledge graph that stores both general knowledge and user-specific wearable data. The retrieval fuses 1) global, long-term relationships estimated with a hierarchical Bayesian model that blends domain knowledge, population patterns and individual data with 2) local, short-term signals that stress anomalies. Experiments are done based on several wearable datasets to prove the improvement on response quality for WAG. Also ablation study is designed to prove the necessity of all modules.

**Strengths:**

+ A benchmark of 10k+ data-linked queries synthesized from multiple wearable datasets to evaluate retrieval and reasoning.

+ From the perspective of originality, this paper is, as of my memory, this paper is the first one doing retrieval on KG-based wearable sensor data. Also the idea of combining global relationships with local signals is also interesting

+ Experiments and ablations are clear.

+ The method and pipeline are presented very clearly.

**Weaknesses:**

1.	Queries are generated from existing datasets to simulate the usage, it probably not reflect real user intents. Is it possible to collect and test on real questions?
2.	The default edge weight is generated by LLM, also the same as KG creation. I doubt weather is could import bias. Comparison should be made to prove the effectiveness of using LLM for KG construction.
3.	Efficiency is under reported. Retrieval latency, PKG size, end-to-end token/compute savings should be included in the experiments.
4.	The personalization benefits are not clearly proved. While HBM fuses prior/population/individual, the per-user lift vs. a non-personalized graph isn’t clearly quantified beyond average ranks. You may add additional performance report by user data length (from cold-start to sufficient user data)
5.	Baselines are not sufficient, why not using GraphRAG(think-on-graph, Graph-constrained Reasoning, HippoRAG, etc.)- as baselines since your retrieval is base on KG?

**Questions:**

1.	Same at 1st point in weakness. Is it possible to collect and test on real questions?
2.	Why use LLM to generate default edge weight? Why not data-driven method?
3.	Edges are undirected. Did you try directed or typed edges, and if not, why?
4.	Could you try to add extra experiments on 1) purely data-driven graphs as KG alternatives and 2) more graph-based RAG baselinew?

**Details Of Ethics Concerns:**

Publicly available multimodal lifelogging datasets are used.

---

### Official Review · Reviewer_7VMx · 2025-10-31

**Soundness:** 1
**Presentation:** 1
**Contribution:** 2
**Rating:** 2
**Confidence:** 3

**Summary:**

This paper introduces Wearable As Graph (WAG), a framework for automating context retrieval from multimodal wearable sensor data using knowledge graphs between different health metrics. The key goal is to enable LLM-based health analysis that is contextually grounded and individualized without requiring manual selection of specific sensor data. The authors propose to have an LLM determine the relevancy and connections between different health metrics to create the knowledge graph. During retrieval the KG is used to select the relevant metrics via a combination of LLM

The authors introduce a pipeline of creating data-associated queries with health metrics curated from  existing wearable datasets. Using both LLM- and human-based evaluations across criteria such as insightfulness, relevance, groundedness, and personalization, the authors show WAG outperforms baseline and RAG methods, achieving a ~70% win rate.

**Strengths:**

S1. The idea of using a knowledge graph to select contextually relevant metrics is interesting and potentially useful.

S2. The authors generated a large set of queries that would likely be useful for evaluation. Introducing the notion of query openness as a dimension to study performance on different queries could be very insightful into model behavior.

**Weaknesses:**

W1. While the idea of personalized knowledge graph and graph-based retrieval is conceptually interesting, it is hard to understand fundamentally why we need to create this structure. There are 52 metrics that we need to determine are relevant for a query and it seems WAG does not use the actual metric data (just metric description and web search results) to construct this graph. If its just 52 metric names and description could a good LLM not determine what the most relevant metrics are by just prompting it with all 52 metrics? At the very least this should be a very reasonable baseline. The lack of detail in the existing baselines also make it difficult to assess the validity of WAG (W3).

W2. Aspects of the framework lack either validity and/or motivation why a certain design choice was made. For example, if the goal is to have data that is representative and diverse (line 131), why only focus on those with high data completeness (which may not be representative of real-world wearable data)? What were the reasons behind openness categories? When asking LLMs to define openness, were these aligned with what humans would have given? What makes a LLM defined edge weight valid? How were the relationship scoring categories defined?

W3. Overall, the presentation of the paper and details for reproducibility could be improved. The three baselines are introduced in line 409 with very minor detail. Even looking at the qualitative examples in Appendix J and prompts in Appendix K, it is unclear how the baselines exactly work. What is the prompt for the RAG condition? What is the retriever? How are items indexed? Why does the baseline context include the metric data but the others do not in Appendix J?

It would be helpful to have an end-to-end example for each. Likewise, details such as the llm used (DeepSeek-V3) is deep in the appendix and we don't know details such as the temperature being used. Without these details it is hard to understand/access the validity of the paper.

**Questions:**

See above.

---

### Official Review · Reviewer_C5cQ · 2025-10-31

**Soundness:** 3
**Presentation:** 2
**Contribution:** 3
**Rating:** 4
**Confidence:** 2

**Summary:**

The paper introduces WAG, which is a graph-based RAG framework that enables LLMs to analyze multimodal wearable data. The method builds personalized KGs combining population and user leveling modeling via a hierarchical Bayesian model, which allows dynamic retrieval of relevant context for user query. The paper also created a benchmark with 10k queries from public wearable datasets, and experiments show significant improvements over baselines.

**Strengths:**

1. I think the scope of the pipeline provided by the paper is novel and important, as highlighted in the introduction section.

2. Integration of hierarchical Bayesian modeling for combining population and individual knowledge is elegant and looks theoretically good to me.

3. The paper evaluates using both LLM as judge and human raters, which is robust to me.

**Weaknesses:**

1. Figure 1 needs more explanation in its caption. It's a bit hard to understand the context by directly reading the figure.

2. In section 4, why the prior distribution is an Gaussian? How to initialize the weights mu^{prior}? I don't see it in Algo 1 or Algo 2.

3. It seems that there is no comparison with some embedding-based retrieval baselines such as GraphRAG in the experiments session.

4. How does the latency look like for this pipeline? It seems that the work combines efforts from LLMs and Bayesian models,

5. I feel in section 4, it's very confusing regarding the definition of W_{pop} and w_{ind}. What are their relationship with R_x^{pop} and R_x^{s}? What are V_x^{s} and V_x^{pop}.

**Questions:**

Please refer to my comments in weaknesses.

---

### Official Review · Reviewer_DoTW · 2025-11-03

**Soundness:** 2
**Presentation:** 2
**Contribution:** 2
**Rating:** 4
**Confidence:** 4

**Summary:**

This paper addresses the challenge of applying Large Language Models (LLMs) to the vast, long-term, and multimodal data generated by wearable sensors. To solve this, they introduce Wearable As Graph (WAG), a framework that automates context retrieval for LLMs using personalized knowledge graphs (PKGs). WAG constructs a base knowledge graph mapping relationships between wearable modalities and then personalizes it by integrating user-specific data. The core contribution is a dynamic, data-driven retrieval pipeline that identifies the most relevant data nodes to provide to the LLM based on a user's query.

**Strengths:**

1. The overall paper presentation is clear and easy to follow.
2. The core strength is the WAG retrieval algorithm. It moves beyond simple semantic RAG by incorporating a deeply personalized, Bayesian-informed weighting ($w^{global}$) with a context-aware, dynamic weighting ($w^{local}$).
3. The overall vision of a personalized knowledge graph (PKG) that adapts to users is an intuitively appealing goal.

**Weaknesses:**

1. In terms of the evaluation, the benchmark is synthetic (LLM-generated queries) and the evaluation is circular (LLM-as-judge). The 70% win rate is therefore not a meaningful metric of real-world performance.
2. The overall novelty and the contribution are relatively incremental. The paper is more like converting the traditional building-KG-and-retrieving framework to the wearable sensors data, which have been largely explored in the previous studies.
3. The compared baselins are rather simple, where only base and rag methods are compared, which do not pose much applicability for the evaluation results.

**Questions:**

1. Given that the query set was generated by an LLM and judged by an LLM, how can you disprove the null hypothesis that the 70% win-rate is merely an artifact of this closed-loop system rewarding its own internal logic?2
2. Why should the low IRR from human evaluators  be interpreted as a "challenge" rather than a direct invalidation of the evaluation task and the LLM-as-judge's results?
3. How does the system handle known non-monotonic health relationships (e.g., the U-shaped curve for sleep duration and health) when the HBM is limited to monotonic Spearman correlations?

---

### Note · Authors · 2025-11-21

I have read and agree with the venue's withdrawal policy on behalf of myself and my co-authors.